# Systematic review and network meta-analysis comparing antithrombotic agents for the prevention of stroke and major bleeding in patients with atrial fibrillation

Chris Cameron,[1,2] Doug Coyle,[1] Trevor Richter,[3] Shannon Kelly,[2] Kasandra Gauthier,[3] Sabine Steiner,[4] Marc Carrier,[5] Kathryn Coyle,[6] Annie Bai,[3] Kristen Moulton,[3] Tammy Clifford,[1,3] George Wells[1,2]

▶ Prepublication history and additional material is available. To view please visit the journal (http://dx.doi.org/10.1136/bmjopen-2013-004301).

For numbered affiliations see end of article.

**Correspondence to**
Dr George Wells;
gawells@ottawaheart.ca

## ABSTRACT

**Objective:** To examine the comparative efficacy and safety of antithrombotic treatments (apixaban, dabigatran, edoxaban, rivaroxaban and vitamin K antagonists (VKA) at a standard adjusted dose (target international normalised ratio 2.0–3.0), acetylsalicylic acid (ASA), ASA and clopidogrel) for non-valvular atrial fibrillation and among subpopulations.

**Design:** Systematic review and network meta-analysis.

**Data sources:** A systematic literature search strategy was designed and carried out using MEDLINE, EMBASE, the Cochrane Register of Controlled Trials and the grey literature including the websites of regulatory agencies and health technology assessment organisations for trials published in English from 1988 to January 2014.

**Eligibility criteria for selecting studies:** Randomised controlled trials were selected for inclusion if they were published in English, included at least one antithrombotic treatment and involved patients with non-valvular atrial fibrillation eligible to receive anticoagulant therapy.

**Results:** For stroke or systemic embolism, dabigatran 150 mg and apixaban twice daily were associated with reductions relative to standard adjusted dose VKA, whereas low-dose ASA and the combination of clopidogrel plus low-dose ASA were associated with increases. Absolute risk reductions ranged from 6 fewer events per 1000 patients treated for dabigatran 150 mg twice daily to 15 more events for clopidogrel plus ASA. For major bleeding, edoxaban 30 mg daily, apixaban, edoxaban 60 mg daily and dabigatran 110 mg twice daily were associated with reductions compared to standard adjusted dose VKA. Absolute risk reductions with these agents ranged from 18 fewer per 1000 patients treated each year for edoxaban 30 mg daily to 24 more for medium dose ASA.

**Conclusions:** Compared with standard adjusted dose VKA, new oral anticoagulants were associated with modest reductions in the absolute risk of stroke and major bleeding. People on antiplatelet drugs experienced more strokes compared with anticoagulant drugs without any reduction in bleeding risk. To fully elucidate the comparative benefits and harms of antithrombotic agents across the various subpopulations, rigorously conducted comparative studies or network meta-regression analyses of patient-level data are required.

**Systematic review registration number:**
PROSPERO registry—CRD42012002721.

## Strengths and limitations of this study

- This network meta-analysis includes data on edoxaban from the recently published ENGAGE AF-TIMI 48 trial.
- The studies pooled in this meta-analysis include antiplatelet agents in addition to newer oral anticoagulants and standard adjusted dose vitamin K antagonists.
- We present findings on the relative and absolute scale, and graphically illustrate absolute risks using icon arrays.
- We report detailed a priori subgroup analyses for stroke or systemic embolism and major bleeding using data derived from FDA Public Summary Documents.
- Comparisons between antithrombotic treatments were largely comprised of single, albeit large, studies.
- There is notable heterogeneity and the small number of studies limits the analyses that can be conducted to account for heterogeneity in the absence of patient-level data.
- Icon arrays only consider the primary efficacy and safety endpoints of underlying studies and do not account for the different clinical consequences associated with each of the outcomes.

## INTRODUCTION

Atrial fibrillation (AF) is a common cardiac arrhythmia associated with increased morbidity and mortality.[1] [2] Patients with AF sustain an increased risk of arterial thromboembolism and stroke, which are associated with high recurrence and substantial debilitating impact. Therefore, antithrombotic strategies using anticoagulant drugs and antiplatelet agents are recommended for patients with AF presenting with risk factors for stroke. Antithrombotic therapy is also associated with a risk of bleeding; therefore, the beneficial effects on stroke prevention should always be compared against a patient's risk of major bleeding.[1] [2]

Existing guidelines recommend anticoagulant therapy for patients at intermediate or high risk of stroke.[1] [2] Although standard adjusted dose vitamin K antagonist (VKA) (eg, warfarin) has been the cornerstone treatment for reducing the risk of stroke or systemic embolism (SE) in this population, it is associated with several drawbacks[2] which have prompted the development of newer oral anticoagulants such as direct thrombin (dabigatran) and Xa (eg, apixaban, edoxaban, rivaroxaban) inhibitors.[1] [2] While some of these newer treatment options have been demonstrated to be efficacious in preventing stroke or SE compared to standard adjusted dose VKA, the relative efficacy and risk of bleeding of newer oral anticoagulants, both among themselves and in comparison to antiplatelet agents, is not clear, especially in certain subpopulations of patients, for example, CHADS$_2$ score <2 or ≥2; time in therapeutic range (TTR) <66% or ≥66%; Age <75 or ≥75 years. Therefore, the objective of this paper was to compare antithrombotic agents for the prevention of stroke and major bleeding in patients with non-valvular AF and among subpopulations.

## METHODS

The published literature was identified by searching MEDLINE, MEDLINE In-Process and Other Non-Indexed Citations. EMBASE, BIOSIS Previews, PubMed and the Cochrane Central Register of Controlled Trials were searched through the Ovid interface to identify English-language clinical articles published from 1988 to 23 January 2014. Regular alerts were also established. Websites of regulatory agencies, Canadian and major international health technology assessment agencies, clinical practice guidelines as well as The Cochrane Library (2012, Issue 6) and University of York Centre for Reviews and Dissemination databases were searched. Complete details of the electronic search strategy, including any limits used, are reported in online supplementary appendix 1. The protocol was published online and was registered in the PROSPERO International Prospective Register of Systematic Reviews (PROSPERO registry—CRD42012002721).

The population of interest was individuals with non-valvular AF requiring anticoagulation (including all risk levels and regardless of any comorbidities). The following treatments were included in the review: apixaban, dabigatran, edoxaban, rivaroxaban, standard adjusted dose VKA, acetylsalicylic acid (ASA), ASA and clopidogrel. Numerous outcomes were considered and results are reported elsewhere[3]; however, the focus of this publication is the outcomes of all-cause stroke or SE and major bleeding as defined by the International Society on Thrombosis and Haemostasis (ISTH) definition),[4] which were the primary efficacy and safety outcomes used in the newer oral anticoagulant studies.

Active and placebo-controlled randomised controlled trials (RCTs) were selected for inclusion if they were published in English, included at least one antithrombotic treatment under review (using pre-specified doses), reported data for stroke/SE or major bleeding, and involved patients with non-valvular AF eligible to receive anticoagulant therapy, regardless of the level of stroke risk. Trials that included patients with contraindication to anticoagulant treatment were excluded. VKA trials were included if the dose was adjusted to a target international normalised ratio (INR) 2.0–3.0. Any dose of ASA was considered for inclusion, but ASA dose was stratified in the analysis as low (≤100 mg daily), medium (>100 mg to ≤300 mg daily) or high (>300 mg daily). We only included new oral anticoagulants which had at least one large phase III study. Three reviewers independently extracted the data on baseline characteristics, intervention(s) evaluated, including dose, duration and relevant co-medication, and results for each included article, using a standardised template. All extracted data were checked for accuracy by three independent reviewers. Any disagreements in the assessment of these data were resolved through discussion until consensus was reached. Quality assessment of RCTs was also performed independently by two reviewers using a standardised table based on major items from the SIGN 50 instrument for internal validity.[5] The trial selection process is presented in a flow chart based on the Preferred Reporting Items for Systematic Reviews and Meta-Analyses (PRISMA)[6] statement (see online supplementary appendix 2).

Bayesian network meta-analyses were conducted using WinBUGS software (MRC Biostatistics Unit, Cambridge, UK). A binomial likelihood model[7] which accounts for use of multi-arm trials was used for analyses, given outcomes were dichotomous and included multi-arm trials. Trials with zero cells in both arms or nodes, where there were no events, were excluded from the evidence networks because they do not contribute information or allow interpretable information.[7] Both fixed and random-effects network meta-analyses were conducted, although the fixed-effects model was chosen for the reference case analysis, as the evidence network was largely comprised of single study connections. We modelled point estimates and 95% credible intervals for OR using Markov Chain Monte Carlo methods. The absolute risk difference per 1000 patients treated each year for each

outcome was also calculated, based on the standard adjusted dose VKA arm of the Randomised Evaluation of Long-Term Anti-coagulation Therapy (RE-LY) trial.[8] The RE-LY trial[8] was selected because it contained data for both CHADS$_2$ <2 and CHADS$_2$ ≥2 and had detailed data available from the Food and Drug Administration (FDA) Public Summary Report.[9] We also assessed the probability that each treatment was the most efficacious regimen, the second best, the third best and so on.[10] Vague or flat priors, such as N (0, 100$^2$), were assigned for basic parameters throughout, although informative priors were considered.[11] Assessment of model fit was based on the deviance information criterion (DIC) and comparison of residual deviance to number of unconstrained data points.[7] [12] To ensure that convergence was reached, trace plots and the Brooks-Gelman-Rubin statistic were assessed.[13] Three chains were fitted in WinBUGS for each analysis, with at least 40 000 iterations, and a burn-in of at least 40 000 iterations. We also qualitatively compared the results from our network meta-analysis with direct pairwise estimates. Frequentist pairwise meta-analyses were conducted using R (R Language and Environment for Statistical Computing, Vienna, Austria) and package meta.[14]

A key assumption behind network meta-analysis is that the analysed network is consistent, that is, there is no conflict between direct and indirect evidence.[15] To assess inconsistency, we compared deviance and DIC statistics in fitted consistency and inconsistency models.[15] We also plotted the posterior mean deviance of the individual data points in the inconsistency model against their posterior mean deviance in the consistency model to identify any loops where inconsistency was present (see online supplementary appendix 11).[15] Additionally, the results from our network meta-analysis were qualitatively compared with direct frequentist pairwise estimates.

Network meta-analysis also requires that studies are sufficiently similar in order to pool their results.[16] Available study and patient characteristics were assessed to ensure similarity and to investigate the potential impact of heterogeneity on effect estimates. Clinical and methodological heterogeneity was identified in a number of areas. The issues identified were similar to those reported in previous systematic reviews of anticoagulant drugs (eg, differences in TTR, CHADS$_2$ score).[17] Heterogeneity was assessed by conducting network meta-analysis with the following pre-specified subgroup data reported in the individual RCTs: TTR <66% or ≥66%; CHADS$_2$ score <2 or ≥2; Age <75 or ≥75 years.[17] The data to conduct these subgroup analyses were derived from FDA Public Summary Documents when not available in the publications.[9] [18] [19] Secondary evidence networks were constructed for each of the subgroup analyses. The WinBUGS code[7] and data needed to replicate all analyses are available online.

## RESULTS

### Study characteristics of included studies

The systematic review included 16 individual RCTs (reported in 32 publications and FDA reports); all evaluated the efficacy and safety of antithrombotic agents: apixaban (5 mg twice daily), dabigatran (150 or 110 mg twice daily), edoxaban (30 or 60 mg daily), rivaroxaban (20 mg daily), standard adjusted dose VKA, ASA (low dose (<100 mg daily), medium dose (100–300 mg daily)), or low-dose ASA plus clopidogrel (75 mg daily) in patients with non-valvular AF. None of the included studies directly compared one new oral anticoagulant drug with another (see online supplementary appendix 4). Of the 82 396 randomised patients included in the primary analysis, five large multicentre trials account for 78 296 patients (96%): ENGAGE AF-TIMI 48 (n=21 105),[20] ARISTOTLE (n=18 201),[21] RE-LY (n=18 113),[8] ROCKET-AF (n=14 264)[22] and ACTIVE-W (n=6706).[23] Most trials included stroke and SE as a primary efficacy outcome, while bleeding events were a frequent safety outcome. Follow-up ranged from 12 weeks to 3.5 years. All trials published results between 1989 and 2013. The mean CHADS$_2$ score was reported in eight trials, encompassing the vast majority of patients included in the systematic review. Reported CHADS$_2$ scores were consistent with a high-risk population (CHADS$_2$ ≥2), with patients from ROCKET-AF[22] showing the highest risk for stroke (mean CHADS$_2$=3.5).

Mean age across the included trials ranged from 62 to 83 years (see online supplementary appendix 3). All trials included patients of both gender and had a higher proportion of male participants in most studies. However, the proportions of patients with a prior stroke or TIA varied substantially across the included trials, ranging from 3% in JAST to 55% in ROCKET-AF.[22] The same observation can be made regarding the proportions of patients with other concomitant conditions, that is, congestive heart failure, hypertension, diabetes, myocardial infarction and prior VKA experience. TTR was reported in all but one trial that included a standard adjusted dose VKA treatment arm. Patient baseline characteristics are summarised in online supplementary appendix 3.

The studies were critically appraised individually and the details are reported elsewhere.[3] Overall, there was substantial variation in study quality. However, large multicentre trials comparing newer anticoagulants with standard adjusted dose VKA, such as ENGAGE AF-TIMI 48 (n=21 105), ARISTOTLE (n=18 201), RE-LY (n=18 113) and ROCKET-AF (n=14 264), which account for the vast majority of patients included in the systematic review, appear to be methodologically rigorous.

### Stroke or SE

The evidence network (figure 1) for the primary analysis was comprised of 12 RCTs representing eight treatments in addition to placebo/observation (N=82 396). Four RCTs

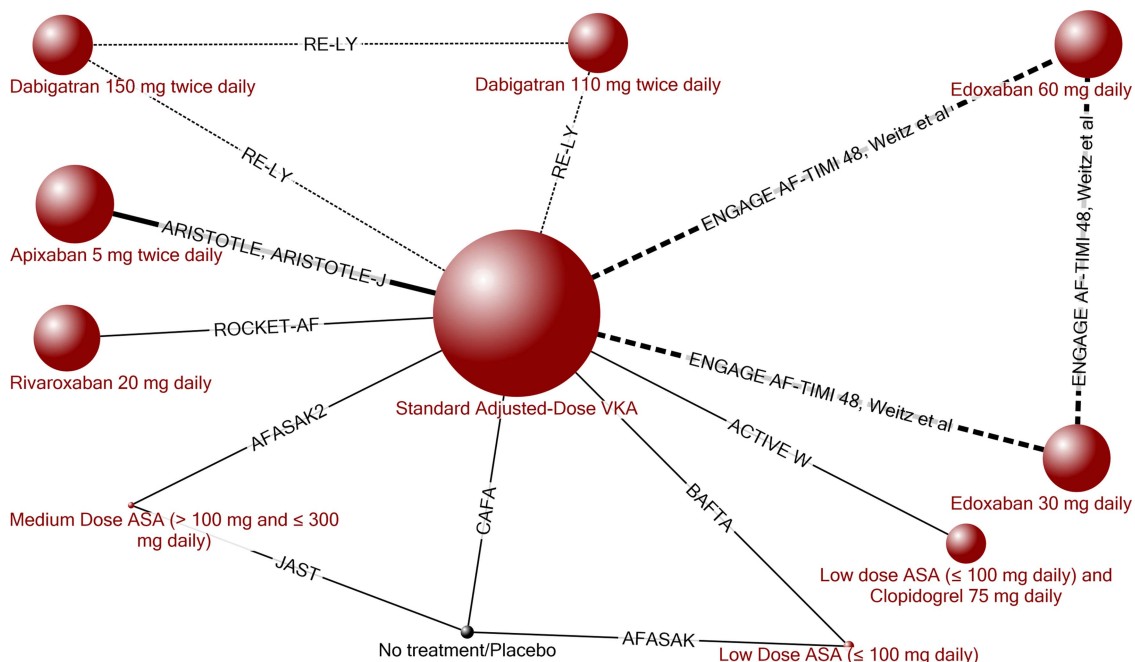

**Figure 1** Evidence network for all-cause stroke or systemic embolism. (Four RCTs, PETRO,[24] WASPO[25] Chung et al[26] and Yamashita et al,[27] were not included in the analysis because they did not report data for this outcome or had zeros in both arms.) The width of the lines is proportional to the number of randomised controlled trials comparing each pair of treatments, and the size of each treatment node is proportional to the number of randomised participants (sample size). A dotted line indicates a three-arm randomised controlled trial and a black node indicates a node included in the analysis but not reported in the main text. ASA, acetylsalicylic acid; RCT, randomised controlled trial; VKA, vitamin K antagonist.

(Chung et al,[26] PETRO,[24] WASPO[25] and Yamashita et al[27]) were not included in the analysis because they did not report data for this outcome or had zeros in both arms.

Dabigatran (150 mg twice daily) and apixaban were associated with reductions in stroke or SE relative to standard adjusted dose VKA (figure 2). The use of these two agents led to absolute risk reductions ranging from 4 to 6 fewer events per 1000 patients treated each year. In contrast, low-dose ASA and the combination of clopidogrel plus low-dose ASA appeared to have a higher risk of stroke or SE than standard adjusted dose VKA, leading to an increase in the number of stroke or SE ranging from 14 to 15 more events per 1000 patients treated each year (figure 4 and see online supplementary appendix 6). No differences were detected between standard adjusted dose VKA and each of the following interventions: dabigatran (110 mg twice daily), edoxaban (30 mg or 60 mg daily), rivaroxaban and medium-dose ASA (figure 3).

The estimates of effect derived from the direct pairwise comparisons aligned well with those obtained from the network meta-analysis in direction and magnitude (see online supplementary appendix 6). Furthermore, the posterior mean residual deviance (29.41) is close to the number of unconstrained data points (27), which is an indication of reasonable model fit. A number of subgroup analyses and alternative modelling strategies were conducted on the primary analysis, including adjusting for $CHADS_2$, TTR and age (table 1). We also conducted a sensitivity analysis where we included the AVERROES[28]

and ACTIVE A[29] trials (see online supplementary appendix 12). These studies were excluded from the main analysis because we required patients to be eligible for anticoagulant therapy, including treatment with a VKA. The results differed slightly from the reference case (see online supplementary appendix 12). The primary analysis was conducted using a fixed-effects model because the network was comprised of single study connections; these results were also compared against those obtained using a random-effects model with informative and vague priors on the SD and found to be similar in magnitude, although wider credible intervals were observed (see online supplementary appendix 9).

### Major bleeding

The evidence network for the primary analysis for major bleeding was comprised of 15 RCTs representing eight treatments in addition to placebo/observation (N=83 015). The evidence network for major bleeding is similar to the evidence network for stroke or SE (figure 1) but includes one extra RCT (WASPO[25]) comparing medium-dose ASA with standard adjusted dose VKA, and two extra RCTs (Chung et al,[26] and Yamashita et al[27]) comparing edoxaban with standard adjusted dose VKA (see online supplementary appendix 5).

Edoxaban 30 mg daily, apixaban, edoxaban 60 mg daily and dabigatran 110 mg twice daily were associated with reductions in the risk of major bleeding compared with standard adjusted dose VKA (figure 2). No differences for major bleeding were detected between

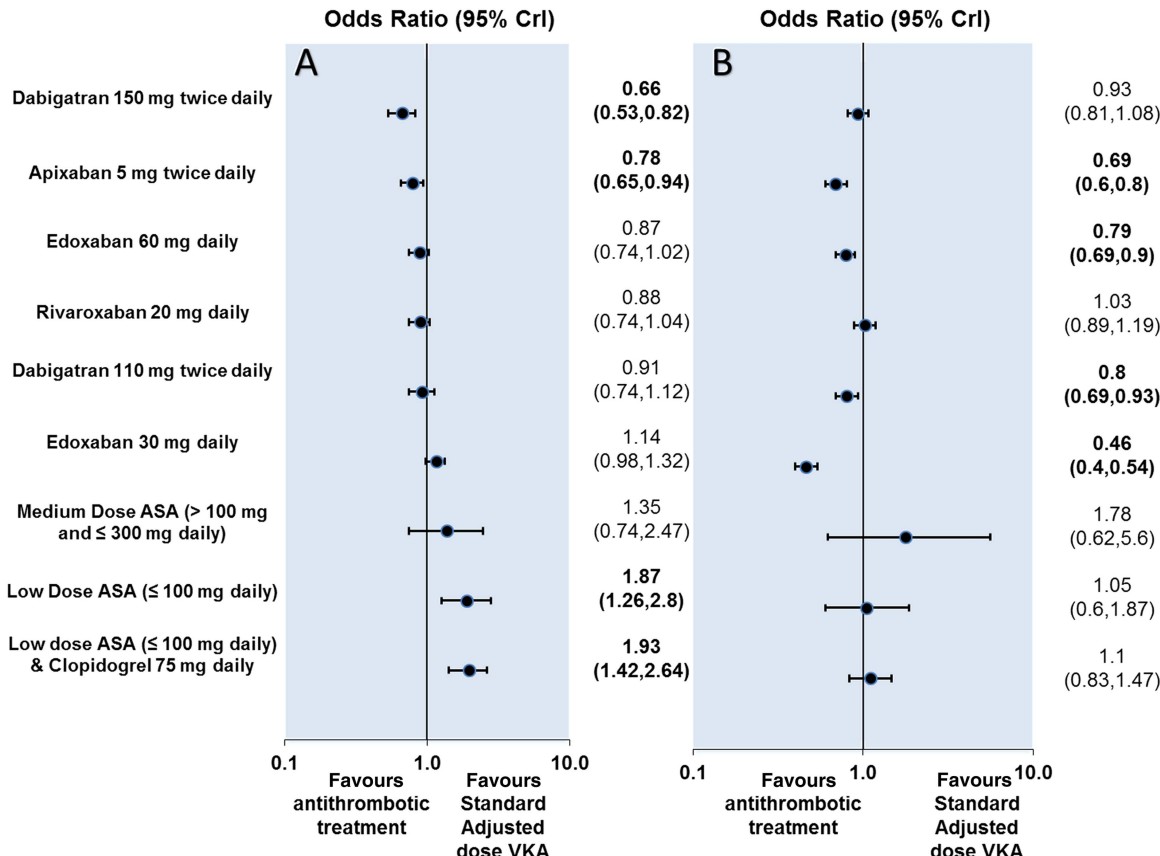

**Figure 2** OR for all-cause stroke or systemic embolism (A) and major bleeding (B) in Bayesian network meta-analysis versus standard adjusted dose VKA. CrI, credible interval; VKA, vitamin K antagonist.

standard adjusted dose VKA and each of the remaining interventions: dabigatran 150 mg twice daily, rivaroxaban, clopidogrel plus low-dose ASA and all ASA dosages (figure 2). The absolute risk difference of major bleeding relative to standard adjusted dose VKA ranged from 18 fewer to 24 more events per 1000 patients treated per year (figure 4 and see online supplementary appendix 7). A summary of the results for the Bayesian and direct pairwise meta-analyses is shown in online supplementary appendix 7, and Bayesian network meta-analysis results are represented graphically in figure 2. Complete results from the Bayesian network meta-analysis for all possible comparisons are presented in figure 3.

The estimates of effect derived from the direct pairwise comparisons aligned well with those obtained from the network meta-analysis in direction and magnitude (see online supplementary appendix 7). Furthermore, the posterior mean residual deviance (38.0) is close to the number of unconstrained data points (35), which is an indication of reasonable model fit. A number of subgroup analyses and alternative modelling strategies were conducted on the primary analysis (table 1). We also conducted a sensitivity analysis where we included the AVERROES[28] and ACTIVE A[29] trial (see online supplementary appendix 12), as well as analyses using a random-effects model with vague and informative priors (see online supplementary appendix 9).

### Benefit–harm assessment—stroke or SE versus major bleeding

Figure 3 summarises the results for all pairwise comparisons derived from the Bayesian fixed-effects network meta-analysis (see online supplementary appendix 9). Results relative to standard adjusted dose VKA have been discussed above. For pairwise comparisons among newer oral anticoagulants for stroke or SE, dabigatran 150 mg twice daily was associated with fewer events versus dabigatran 110 mg twice daily, edoxaban 30 mg daily, edoxaban 60 mg daily and rivaroxaban. Apixaban and rivaroxaban were also associated with fewer events compared to edoxaban 30 mg daily. There were no differences among low-dose ASA, medium-dose ASA, and clopidogrel plus low-dose ASA. However, low-dose ASA and the combination of clopidogrel plus low-dose ASA were associated with increases in stroke or SE compared to all newer oral anticoagulants. For major bleeding, apixaban and dabigatran 110 mg twice daily were associated with fewer events versus dabigatran 150 mg twice daily and rivaroxaban. Edoxaban 30 mg daily was associated with fewer events than other new oral anticoagulants, while edoxaban 60 mg daily was associated with fewer events compared with rivaroxaban and clopidogrel plus low-dose ASA. The risk of major bleeding for apixaban was lower compared to clopidogrel plus low-dose ASA. There were no differences associated with the ASA

| Standard Adjusted-Dose VKA | 0.93 (0.81,1.08) | **0.69 (0.6,0.8)** | **0.79 (0.69,0.9)** | 1.03 (0.89,1.19) | **0.8 (0.69,0.93)** | **0.46 (0.4,0.54)** | 1.78 (0.62,5.6) | 1.05 (0.6,1.87) | 1.1 (0.83,1.47) |
|---|---|---|---|---|---|---|---|---|---|
| **0.66 (0.53,0.82)** | Dabigatran 150 mg twice daily | **0.74 (0.6,0.91)** | 0.84 (0.69,1.03) | 1.1 (0.9,1.35) | 0.86 (0.74,1.003) | **0.5 (0.4,0.61)** | 1.91 (0.66,6.08) | 1.13 (0.63,2.04) | 1.18 (0.85,1.63) |
| **0.78 (0.65,0.94)** | 1.19 (0.89,1.58) | Apixaban 5 mg twice daily | 1.14 (0.93,1.38) | **1.49 (1.21,1.82)** | 1.16 (0.94,1.43) | **0.67 (0.54,0.83)** | 2.57 (0.89,8.18) | 1.52 (0.85,2.75) | **1.59 (1.16,2.2)** |
| 0.87 (0.74,1.02) | **1.31 (1.002,1.73)** | 1.11 (0.87,1.41) | Edoxaban 60 mg daily | **1.31 (1.08,1.59)** | 1.02 (0.83,1.25) | **0.59 (0.5,0.69)** | 2.27 (0.78,7.17) | 1.34 (0.75,2.41) | **1.4 (1.02,1.92)** |
| 0.88 (0.74,1.04) | **1.33 (1.01,1.76)** | 1.12 (0.87,1.43) | 1.01 (0.8,1.28) | Rivaroxaban 20 mg daily | **0.78 (0.63,0.96)** | **0.45 (0.37,0.56)** | 1.74 (0.6,5.48) | 1.02 (0.57,1.85) | 1.07 (0.78,1.48) |
| 0.91 (0.74,1.12) | **1.38 (1.11,1.74)** | 1.17 (0.88,1.53) | 1.05 (0.81,1.37) | 1.04 (0.8,1.36) | Dabigatran 110 mg twice daily | **0.58 (0.47,0.72)** | 2.22 (0.77,7.03) | 1.31 (0.73,2.37) | 1.37 (0.99,1.9) |
| 1.14 (0.98,1.32) | **1.73 (1.32,2.26)** | **1.45 (1.15,1.84)** | **1.31 (1.13,1.54)** | **1.3 (1.04,1.63)** | 1.25 (0.97,1.61) | Edoxaban 30 mg daily | **3.85 (1.32,12.18)** | **2.27 (1.26,4.1)** | **2.38 (1.72,3.3)** |
| 1.35 (0.74,2.47) | **2.04 (1.08,3.9)** | 1.72 (0.92,3.24) | 1.56 (0.84,2.91) | 1.54 (0.83,2.88) | 1.48 (0.79,2.8) | 1.18 (0.64,2.21) | Medium Dose ASA (> 100 mg and ≤ 300 mg daily) | 0.59 (0.17,1.94) | 0.62 (0.19,1.84) |
| **1.87 (1.26,2.8)** | **2.84 (1.81,4.5)** | **2.39 (1.55,3.72)** | **2.16 (1.42,3.34)** | **2.14 (1.4,3.31)** | **2.05 (1.32,3.23)** | **1.64 (1.08,2.53)** | 1.39 (0.74,2.61) | Low Dose ASA (≤ 100 mg daily) | 1.05 (0.55,1.97) |
| **1.93 (1.42,2.64)** | **2.93 (2.01,4.3)** | **2.46 (1.73,3.54)** | **2.23 (1.58,3.17)** | **2.2 (1.55,3.14)** | **2.11 (1.46,3.07)** | **1.69 (1.21,2.4)** | 1.43 (0.73,2.81) | 1.03 (0.62,1.7) | Low dose ASA (≤ 100 mg daily) & Clopidogrel 75 mg daily |

**Figure 3** OR from network meta-analyses for stroke or systemic embolism and major bleeding for all pairwise comparisons. (Results between individual treatments, especially newer oral anticoagulants, should be interpreted with caution, given the limitations associated with using a fixed-effects model. See online supplementary appendix 9 for additional details.) ORs for recurrence of stroke or systemic embolism are below the diagonal (row-defining treatment vs column-defining treatment) and those for major bleeding are above the diagonal and in blue (column-defining treatment vs row-defining treatment). To obtain ORs for comparisons in the opposite direction, reciprocals should be taken (eg, the OR for standard dose warfarin compared with apixaban 5 mg twice daily for stroke or systemic embolism is 1/0.78=1.28). Significant results are in bold. VKA, vitamin K antagonist.

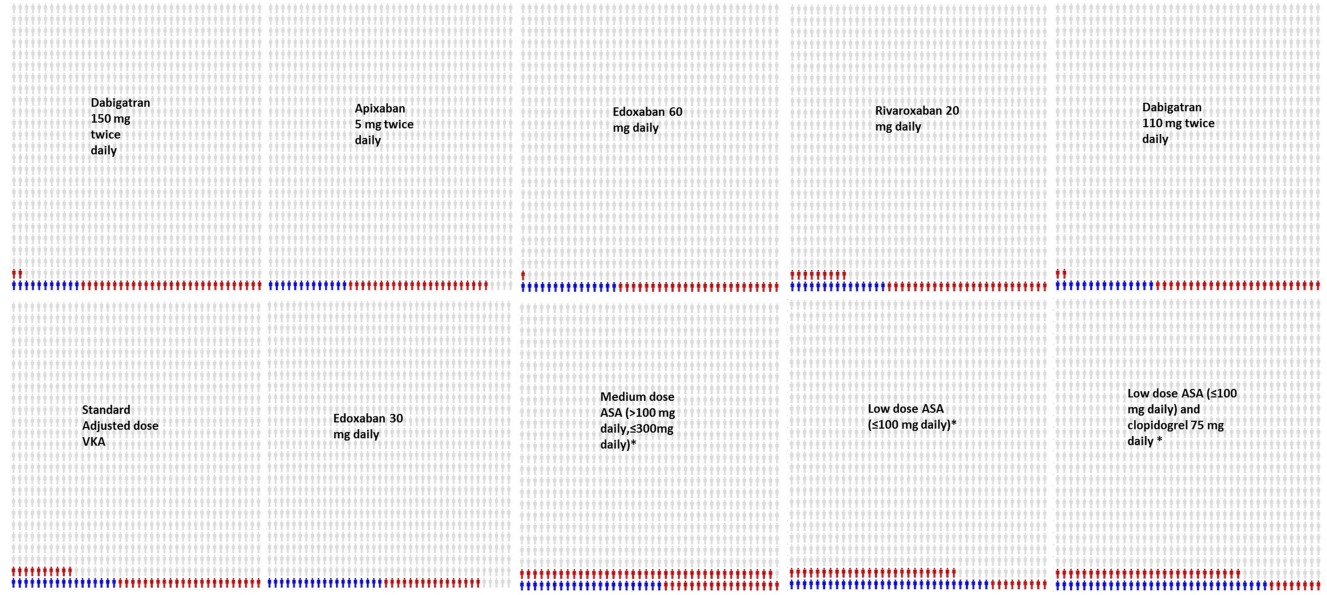

**Figure 4** Icon array illustrating the absolute risks of Stroke or systemic embolism (blue) and major bleeding episodes (red) per 1000 patients treated. (Figures do not reflect uncertainty around effect estimates and there is more uncertainty (ie, wider credible intervals) for low-dose ASA, medium-dose ASA and clopidogrel plus low dose ASA (see online supplementary appendix 8). Underlying studies may also double count haemorrhage stroke.)

**Table 1** Subgroup and sensitivity analyses for stroke or systemic embolism and major bleeding relative to standard adjusted dose VKA, OR±95% credible intervals versus standard adjusted dose VKA

| Treatment | Reference case—fixed effect NMA | Subgroup analysis by CHADS2 score | | Subgroup analysis by age | | Subgroup analysis by TTR | |
|---|---|---|---|---|---|---|---|
| | | CHADS2 <2 | CHADS2 ≥2 | Age <75 years | Age ≥75 years | TTR <66% | TTR ≥66% |
| **Stroke or systemic embolism** | | | | | | | |
| Dabigatran 150 mg twice daily | **0.66 (0.53, 0.82)** | 0.61 (0.37, 0.999) | **0.66 (0.52, 0.85)** | 0.63 (0.46, 0.87) | **0.66 (0.48, 0.90)** | 0.54 (0.39, 0.73) | 0.81 (0.58, 1.11) |
| Apixaban 5 mg twice daily | **0.78 (0.65, 0.94)** | 0.86 (0.57, 1.28) | **0.78 (0.63, 0.95)** | 0.85 (0.67, 1.07) | **0.71 (0.53, 0.95)** | 0.79 (0.62, 1.003) | 0.79 (0.60, 1.05) |
| Edoxaban 60 mg daily | 0.87 (0.74, 1.02) | NA | 0.87 (0.74, 1.01) | 0.91 (0.73, 1.14) | 0.83 (0.66, 1.04) | 0.83 (0.65, 1.07) * | 0.91 (0.73, 1.13)* |
| Rivaroxaban 20 mg daily | 0.88 (0.74, 1.04) | NA | 0.88 (0.74, 1.04) | 0.95 (0.75, 1.2) | 0.80 (0.63, 1.03) | 0.81 (0.65, 1.01) | 0.72 (0.47, 1.10) |
| Dabigatran 110 mg twice daily | 0.91 (0.74, 1.12) | 1.00 (0.64, 1.55) | 0.89 (0.71, 1.12) | 0.93 (0.7, 1.24) | 0.88 (0.66, 1.19) | 0.91 (0.69, 1.19) | 0.90 (0.66, 1.24) |
| Edoxaban 30 mg daily | 1.14 (0.98, 1.32) | NA | 1.14 (0.98, 1.32) | 1.16 (0.94, 1.44) | 1.13 (0.91, 1.40) | 1.01 (0.80, 1.28)* | 1.24 (1.01, 1.24)* |
| Medium-dose ASA (>100 mg and ≤300 mg daily) | 1.35 (0.74, 2.47) | 1.97 (0.63, 6.73)† | NA | NA | NA | NA | NA |
| Low-dose ASA (≤100 mg daily) | **1.87 (1.26, 2.8)** | 2.20 (1.18, 4.25) | 2.10 (0.91, 5.15) ‡ | NA | 2.04 (1.33, 3.20) | NA | NA |
| Clopidogrel 75 mg daily and Low-dose aspirin (≤100 mg daily) | **1.93 (1.42, 2.64)** | 3.16 (1.39, 8.22) | 1.15 (0.84, 1.57) | NA | NA | 1.52 (0.96, 2.45) | **2.40 (1.59, 3.71)** |
| **Major bleeding** | | | | | | | |
| Edoxaban 30 mg daily | **0.46 (0.4, 0.54)** | NA | **0.46 (0.4, 0.54)** | 0.50 (0.41, 0.61)§ | 0.51 (0.41–0.62)§ | 0.45 (0.35, 0.58)*§ | 0.51 (0.43, 0.61)*§ |
| Apixaban 5 mg twice daily | **0.69 (0.6, 0.80)** | 0.59 (0.44, 0.78) | **0.73 (0.62, 0.87)** | **0.73 (0.6, 0.89)** | **0.65 (0.52, 0.80)** | **0.57 (0.45, 0.71)** | **0.81 (0.67, 0.98)** |
| Edoxaban 60 mg daily | **0.79 (0.69, 0.9)** | NA | **0.79 (0.69, 0.90)** | 0.74 (0.62–0.90)§ | 0.82 (0.68–0.98)§ | 0.69 (0.55, 0.87) ast;§ | 0.83 (0.70, 0.97)*§ |
| Dabigatran 110 mg twice daily | **0.80 (0.69, 0.93)** | 0.65 (0.47, 0.89) | 0.86 (0.73, 1.02) | **0.62 (0.49, 0.77)** | 1.02 (0.84, 1.25) | 0.74 (0.61, 0.91) | 0.86 (0.69, 1.06) |
| Dabigatran 150 mg twice daily | 0.93 (0.81, 1.08) | 0.77 (0.57, 1.04) | 1.01 (0.86, 1.19) | **0.70 (0.56, 0.86)** | 1.19 (0.98, 1.45) | 0.76 (0.62, 0.94) | 1.15 (0.94, 1.40) |
| Rivaroxaban 20 mg daily | 1.03 (0.89, 1.19) | NA | 1.03 (0.89, 1.19) | 0.93 (0.76, 1.13) | 1.15 (0.93, 1.41) | 0.92 (0.77, 1.10) | **1.30 (1.01, 1.69)** |
| Low-dose ASA (≤100 mg daily) | 1.05 (0.6, 1.87) | 0.83 (0.42, 1.64) | 1.60 (0.55, 5.02) ‡ | NA | 1.01 (0.57, 1.78) | NA | NA |
| Clopidogrel 75 mg daily and low-dose aspirin (≤100 mg daily) | 1.10 (0.83, 1.47) | 1.51 (0.89, 2.61) | 0.97 (0.69, 1.36)¶ | NA | NA | 0.66 (0.42, 1.02) | 1.66 (1.12, 2.47) |
| Medium-dose ASA (>100 mg and ≤300 mg daily) | 1.79 (0.63, 5.67) | 1.48 (0.11, 19.07)† | NA | NA | NA | NA | NA |

Significant results are in bold.

*Data provided from ENGAGE AF-TIMI 48[20] use TTR >60%.

†Results may be slightly biased against medium-dose ASA. Based on study level results from CAFA and JAST studies. Although these studies consisted primarily of low-risk populations, some patients may have CHADS2 scores greater than 2.

‡Results may be slightly biased against LDASA. Derived from the BAFTA study where CHADS2 subgroup data were stratified by CHADS2 1–2 (vs 0–1) and CHADS2 3–6 (vs 2–6).

§Based on overall time period where overall time period was defined as first dose to the last dose plus 3 days.

¶Estimated from the subgroup study by Healey et al (2008).[29a]

ASA, acetylsalicylic acid; NA, not available; NMA, network meta-analysis; TTR, time in therapeutic range; VKA, vitamin K antagonist.

treatments, both for comparisons among themselves and compared to the anticoagulant treatments.

Figure 4 plots the absolute risk for stroke or SE and major bleeding per 1000 treated for the 10 treatments analysed. Data in figure 4 were obtained from the results presented in online supplementary appendix 6 and 7. Examination of the data in figure 4 suggests that the balance of benefits and harms of the new oral anticoagulants for stroke or SE and major bleeding is positive compared to standard adjusted dose VKA (decrease in stroke or SE and/or major bleeding) and largely similar among one another. Comparison of all anticoagulant treatment (including standard adjusted dose VKA) to ASA or ASA in combination with clopidogrel indicates that whereas the new oral anticoagulants decrease the risk of stroke or SE and major bleeding, ASA has a less favourable balance of benefits and harms, that is, fails to minimise the risk of stroke or SE and/or major bleeding.

## DISCUSSION

We identified 16 RCTs comparing antithrombotic agents for the prevention of all cause stroke or SE and major bleeding in patients with AF. To the best of our knowledge, this is the most up-to-date systematic review and network meta-analysis[30–34] to synthesise the available benefits and harms on newer oral anticoagulants, standard adjusted dose VKA and antiplatelet agents for the prevention of stroke and major bleeding in patients with non-valvular AF (see online supplementary appendix 13). Harenberg et al[30] recently summarised the published indirect comparisons and network meta-analyses evaluating newer oral anticoagulants for AF and an earlier unpublished version of our analysis[17] was considered to be among the most comprehensive analyses at that time.[30] This network meta-analysis expands on that unpublished report.[17] Overall, our findings align with those reported in previously published network meta-analyses[30 31 33]; however, unlike previous analyses,[30] we include recently published edoxaban data,[20] and also include antiplatelet agents. We also present detailed evidence networks[35] to illustrate the body of evidence for each outcome; report analysis using multiple statistical models (see online supplementary appendix 9); report detailed a priori subgroup analyses for each outcome using data derived from FDA Public Summary Documents[9 18]; present findings on the relative and absolute scale,[36 37] and graphically illustrate results using icon arrays. Our network meta-analysis was registered in PROSPERO, and also adheres to PRISMA reporting standards (see online supplementary appendix 10). It also differs from a recently published traditional meta-analysis by Ruff et al,[32] given that we report the comparison of newer oral anticoagulants versus each other and also include comparisons with antiplatelet agents.

For stroke or SE, dabigatran 150 mg and apixaban twice daily were associated with reductions in stroke or SE relative to standard adjusted dose VKA. In contrast, low-dose ASA and the combination of clopidogrel plus low-dose ASA that appeared were associated with increases in stroke or SE relative to standard adjusted dose VKA. No differences were detected between standard adjusted dose VKA and each of the following interventions: dabigatran 110 mg twice daily, edoxaban 30 and 60 mg daily, rivaroxaban, medium-dose ASA and no treatment/placebo. The absolute risk reduction of stroke or SE relative to standard adjusted dose VKA ranged from 6 fewer to 15 more events per 1000 patients treated per year, with dabigatran 150 mg twice daily yielding the largest absolute risk reduction and clopidogrel plus low-dose ASA the largest absolute risk increase.

For major bleeding, edoxaban 30 mg daily, apixaban, edoxaban 60 mg daily and dabigatran 110 mg twice daily were associated with a decreased risk of major bleeding relative to standard adjusted dose VKA. No differences for major bleeding were detected between standard adjusted dose VKA and each of the remaining interventions: dabigatran 150 mg twice daily, rivaroxaban, clopidogrel plus low-dose ASA and all ASA dosages. The absolute risk difference of major bleeding relative to standard adjusted dose VKA ranged from 18 fewer to 24 more events per 1000 patients treated, with edoxaban 30 mg daily yielding the largest absolute risk reduction and medium-dose ASA the largest absolute risk increase. Results of network meta-analyses reflect the relative efficacy and safety profile of antiplatelet and anticoagulant drugs as documented by large clinical trials. Thus, international guidelines[1 2] for prevention of stroke in patients with AF also recommended oral anticoagulation over antiplatelet therapy as the latter had only modest protective effects. With respect to the new oral anticoagulants, none of the guidelines has given a recommendation to prefer one of the new oral anticoagulants over another.[1 2]

Network meta-analysis involves pooling of trials. We observed variability across studies included in the evidence network in terms of CHADS$_2$, age and TTR (see online supplementary appendices 3 and 4). For example, patients using rivaroxaban had a higher risk of stroke because ROCKET-AF[22] included patients with CHADS$_2$ ≥2. As such, we conducted subgroup analysis to adjust for differences in patient and study-level characteristics. We derived data for subgroup analyses from FDA Public Summary documents,[9 18 19] which enhanced our ability to account for variability across studies (eg, use intention to treat population for rivaroxaban versus per-protocol as-treated population reported in publication[22]). Results from these subgroup analyses differed slightly from those reported in the primary analysis, indicating that there may be differences in the benefits and harms across subgroups. This becomes more apparent when subgroup analyses are reported on the absolute scale.[3] It is unclear whether the absolute risk reductions associated with these differences translate into clinically meaningful benefits in practice, especially in subpopulations of patients with a lower baseline risk

of stroke or SE. To fully elucidate the comparative benefits and harms of antithrombotic agents across the various subpopulations, rigorously conducted comparative studies[38 39] or network meta-regression analyses of patient-level data[40] are required.

There are several limitations with the analysis we employed. First, there is notable heterogeneity in patients and study characteristics (see online supplementary appendices 3 and 4). However, the small number of studies limits the analyses that can be conducted to account for heterogeneity in the absence of patient-level data. Second, there is insufficient subgroup data available for other outcomes reported in the RCTs included. As a consequence, detailed analysis was limited to two outcomes—stroke and SE and major bleeding. Third, we reported the results from the fixed-effects model in the main text. We felt that this was appropriate as the nodes in evidence networks are connected largely by single studies. Effect estimates derived from the fixed-effects model aligned more closely with the direct estimates. Reporting results from the fixed-effects model in the main text most likely biases results in favour of the new oral anticoagulants, as treatments that achieved statistically significant results in the primary RCTs retained statistically significant findings when using the fixed-effects model, but not always when using the random-effects model due to the prior[11] on the between-study variance. Nevertheless, the results from the random-effects model have also been reported in online supplementary appendix 9, along with a detailed discussion around the limitations in using the fixed-effects model. Fourth, the absolute risk reduction for the network meta-analysis was calculated using the event rate in the standard adjusted dose VKA arm of the RE-LY study. The RE-LY trial was selected because it contained data for both $CHADS_2 < 2$ and $CHADS_2 \geq 2$ and had detailed data made publicly available following an FDA review of dabigatran. Use of a different study (eg, ARISTOTLE) for baseline event rate data is unlikely to change the results substantially, given that the event rates in standard adjusted dose VKA arms were similar across studies. Fifth, we present findings using graphics that have been largely applied in the network meta-analysis literature,[41] such as a table with OR for all pairwise comparisons. This method of presenting findings on the relative scale is problematic when you are conducting assessments to compare the balance of benefits and harms, which need to be standardised. As such, we also provided results in absolute terms. We report our findings using icon arrays, although it should be noted that these results do not account for the utility values based on patient preferences for each of the outcomes;[42 43] nor do they reflect uncertainty, although we report absolute reductions on the risk benefit plane reflecting uncertainty in online supplementary appendix 8. There are other methods available that can incorporate patient preferences for outcomes.[42 43] Further, estimates of benefits and harms with several of the

therapies (eg, rivaroxaban) come from one trial, and thus such data are not particularly robust. Further research is needed that compares the balance of benefits and harms using other research methodologies, incorporating other relevant outcomes, patient preferences for each of the outcomes[42 43] and additional studies when they become available. Finally, haemorrhagic stroke is also considered a major bleed in underlying studies. Accordingly, we may have double counted haemorrhage stroke in this analysis. Unfortunately, we are not able to account for this issue in all included studies. Nonetheless, it is important to note this limitation and highlight that this potentially biases results in favour of newer oral anticoagulants, given that these agents were associated with reductions in these events.

## CONCLUSIONS

Compared with standard adjusted dose VKA, dabigatran 150 mg and apixaban twice daily were associated with reductions in the incidence of stroke or SE in patients with AF, whereas low-dose ASA and the combination of clopidogrel plus low-dose ASA increased the risk relative to standard adjusted dose VKA at preventing stroke or SE. Edoxaban 30 mg daily, apixaban, edoxaban 60 mg daily and dabigatran 110 mg twice daily were associated with reduced risk in major bleeding compared with standard adjusted dose VKA. Although the results of the current review revealed that there were differences in stroke or SE and major bleeding among antithrombotic agents in the management of patients with AF in relative terms, it is unclear whether the absolute risk reductions associated with these differences translate into clinically meaningful benefits in practice, especially in subpopulations of patients with a lower baseline risk of stroke or SE. Rigorously conducted comparative studies or network meta-regression analyses of patient-level data are required to fully elucidate the comparative benefits and harms of antithrombotic agents across the various subpopulations.

**Author affiliations**
[1]Department of Epidemiology and Community Medicine, University of Ottawa, Ottawa, Canada
[2]University of Ottawa Heart Institute, Ottawa, Canada
[3]Canadian Agency for Drugs and Technologies in Health, Ottawa, Canada
[4]Medical University of Vienna, Vienna, Austria
[5]Thrombosis Program, Division of Hematology, Department of Medicine, University of Ottawa, Ottawa, Canada
[6]Applied Health Economics Research Unit, Ottawa, Canada

**Acknowledgements** The authors would like to acknowledge Carolyn Spry. She was the Information Specialist from CADTH on the review who conducted the search strategy. We would also like to thank Dr Robert Giugliano for providing the subgroup data for edoxaban and his thoughtful comments on earlier drafts of the manuscript during the peer review process.

**Contributors** CC conceived the study design, performed the data extraction and analysis, and drafted the manuscript. DC and GW conceived the study design, helped with the data analysis and reviewed the manuscript for important intellectual content. TR conceived the study design, performed the systematic review and data extraction, and reviewed the manuscript for important intellectual content. SK, SS, KG, AB, KM and KC performed the

systematic review and data extraction, and reviewed the manuscript for important intellectual content. TC, MC and KC conceived the study and reviewed the manuscript for important intellectual content. All authors have read and approved the final version of the manuscript.

**Funding** This research was supported through a grant from the Canadian Institute of Health Research (CIHR) Drug Safety and Effectiveness Network (Funding reference number – 116573) and received financial support from the Canadian Agency for Drugs and Technologies in Health. CC is a recipient of a Vanier Canada Graduate Scholarship through CIHR (Funding reference number – CGV 121171) and has received funding from Canadian Network and Centre for Trials Internationally (CANNeCTIN). He is also a trainee on the CIHR Drug Safety and Effectiveness Network team grant (Funding reference number – 116573). MC is a recipient of a New Investigator Award from the Heart and Stroke Foundation of Canada and holds a T2 research chair in cancer and thrombosis from the University of Ottawa.

**Competing interests** None.

**Provenance and peer review** Not commissioned; externally peer reviewed.

**Data sharing statement** Data available from the Dryad Digital Repository: doi:10.5061/dryad.cf6m2. Additional data can be requested by emailing George Wells: gawells@ottawaheart.ca.

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
