## [Reviewer comments · BMJ Open]

Some articles will have been accepted based in part or entirely on reviews undertaken for other BMJ Group journals. These will be reproduced where possible.

ARTICLE DETAILS

TITLE (PROVISIONAL)	Systematic Review and Network Meta-analysis comparing Antithrombotic Agents for the Prevention of Stroke and Major Bleeding in Patients with Atrial Fibrillation
AUTHORS	Cameron, Chris; Coyle, Douglas; Richter, Trevor; Kelly, Shannon; Gauthier, Kasandra; Steiner, Sabine; Carrier, Marc; Coyle, Kathryn; Bai, Annie; Moulton, Kristen; Clifford, Tammy; Wells, George

VERSION 1 - REVIEW

REVIEWER	Robert Giugliano Brigham and Women's Hospital USA
REVIEW RETURNED	15-Nov-2013

GENERAL COMMENTS	This is manuscript presents a network meta-analysis of clinical trials of antithrombotic therapy in patients with atrial fibrillation. The authors report results from 12 trials in over 60,000 patients and conclude that the new oral anticoagulants (NOACs) modestly reduce stroke and major bleeding compared to Vitamin K antagonists (VKAs), while antiplatelet agents were associated with more strokes (with no reduction in bleeding). The approach is rigorous and conclusions appropriate. While I do believe the authors' findings are valid, there are several major points that deserve consideration, and a number of minor suggestions that could improve this manuscript. The authors should address the following major issues: 1. This approach is not particular novel (contrary to the authors' contention on page 11); they should reference prior similar network analyses (which now exceed 10 in the published literature) and include a new section in the discussion that compares their analyses to the prior similar analyses.2. The authors indicate that the search for publications to include in the paper went through June 8, 2012, which is now nearly 1.5 years ago. The search for new trials to include in this analysis must be updated.3. Why did the authors not include larger phase II trials with NOACs, some of which randomized more than 1000 patients?4. Since data with the largest RCT of a NOAC vs warfarin will be available in November 2013, the authors should include these latest data with edoxaban.5. How did the authors handle data where there were 2 experimental arms compared to the same control (e.g., RE-LY). What methods were used in the modeling and statistical analyses to avoid extra weight assigned to a control group (e.g., warfarin in RE-LY) that
--

served as the same comparator group for 2 of the NOAC therapies (e.g., dabigatran 150 BID and dabigatran 110 BID in RE-LY)?

6. The benefit-harm assessment is an imbalanced one, as the authors have selected “major” bleeding for the harm vs. stroke/SEE for benefit. While some types of “major” bleeding carry a poor prognosis that is similar in severity to a stroke or SEE, others (such as a 2 gm/dL decline in hemoglobin) do not. The authors should repeat this analysis striking a better balance (e.g., use life-threatening bleeding or ICH as the harm vs. ischemic stroke/SEE as the benefit) or assign weighted utilities to these events. Otherwise, this is not really a fair balance of benefit and harm. This becomes a quite obvious limitation when inspecting Figure 4 – placebo/no treatment finishes a close 2nd to apixaban with only 1 more event per thousand (summing stroke + SEE + major bleed).

7. A casual review of Figure 2 might lead the reader to believe that “no treatment/placebo” is not such a bad option, since the confidence bound for benefit overlaps 1.0 (not worse than VKA), and on the safety side, no treatment/placebo is tends to look better than VKA therapy. Yet, this differs from the conclusions of prior VKA vs placebo meta-analyses that are the cornerstone of the guideline recommendations. What explains the difference and how do the authors propose to respond to the argument that no treatment/placebo isn't an equally good option as compared to VKA?

8. In Figure 4, how did the authors account for the fact that haemorrhage stroke is also a major bleed? Were these events double-counted? If not, to which group were they ascribed?

Minor issues (in order of appearance)

1. The wording in the abstract results are a bit confusing and hard to follow with regard to which therapy(ies) are being compared to which. Greater clarity is necessary here so that the review can easily understand the various comparisons being made.

2. Page 1, 2nd paragraph of the introduction. This is all well-described in the literature that includes dozens of meta-analyses (and several network analyses) on this topic. The paragraph could be substantially shortened.

3. Page 3, methods: Please define high, medium, and low dose ranges for aspirin.

4. Page 3, methods, last paragraph: Unclear if the authors excluded all the data from a study because of one data element with a zero cell (which would be undesirable), or just did not include a trial with a zero cell from that specific analysis.

5. Page 4, ROCKET-AF included data stratified by CHADS2 score. The RE-LY are no longer the most recent.

6. The authors should register this analysis online (e.g., PROSPERO or metcardio.org?)

7. The authors should acknowledge that the estimates of benefit and harm with several of the therapies (e.g., rivaroxaban) come from 1 trial, and thus such data are not particularly robust.

8. Figure 1, some of the nodes are either too small or missing (e.g., clopidogrel + aspirin). If too small, then size up all the nodes so that it is possible to see even the node that corresponds to the treatment with the least number of patients. This comment also applies to Appendices 5, 12A, 12B.

9. Figure 2: the size of the point estimate should be proportional to the number of patients (or trials) studied with that therapy.

	10. Figure 4 – I like this figure, but suggest separate rows for the blue and red avatars. That would facilitate a quick comparison of each element (stroke/SEE, major bleeding). It would also dissuade the reader from simply comparing the sum of red+blue across treatments. If one did do that, placebo is 2nd best only to apixaban (and differ by only 1 patient per thousand)! See major comment #7. 11. Appendix 8 is an interesting plot. It could be improved by selecting a type of bleeding that is similar in severity to a stroke/SEE, or by changing the scale of the Y-axis to account for a difference in weighting between major bleeding and stroke/SEE.
--	--

REVIEWER	Goette, Andreas ST:V and OvGU
REVIEW RETURNED	28-Nov-2013

GENERAL COMMENTS	Unfortunately, the ENGAGE AF study was presented during the AHA 2013 in Dallas. The study included more than 21.000 patients. The general term of the presented manuscript is therefore misleading. The authors ust include the data from the ENGAGE AF sudy. In addition, they must show how theri results are related to the latest meta-analysis encompassing all NOAC's (see Ruff et al The Lancet 2013)
---

REVIEWER	Hutton, Jane The University of Warwick, UK.
REVIEW RETURNED	20-Dec-2013

GENERAL COMMENTS	This is a very thorough, careful study and report. Minor points: Page 15, line 22: 'with in' - should this be 'with increasing'? Something is missing. Page 17, line 15 'discussion' not 'discussing' Page 29, ref 34 - The two lower case 'a's are presumably wrong? The first should be 'A', the second I'm not sure. I have one concern: the completion rates for the studies varied from 72% to 92%. The largest study had on 74% completion. Some assessment of whether there was differential completion by drug, and the sub-group factors would be sensible, with a brief comment. I don't think further review is required.
---

VERSION 1 – AUTHOR RESPONSE

Reviewer 1 (Robert Giugliano):

This is manuscript presents a network meta-analysis of clinical trials of antithrombotic therapy in patients with atrial fibrillation. The authors report results from 12 trials in over 60,000 patients and conclude that the new oral anticoagulants (NOACs) modestly reduce stroke and major bleeding compared to Vitamin K antagonists (VKAs), while antiplatelet agents were associated with more strokes (with no reduction in bleeding). The approach is rigorous and conclusions appropriate. While I do believe the authors' findings are valid, there are several major points that deserve consideration,

and a number of minor suggestions that could improve this manuscript.

The authors should address the following major issues:

This approach is not particularly novel (contrary to the authors' contention on page 11); they should reference prior similar network analyses (which now exceed 10 in the published literature) and include a new section in the discussion that compares their analyses to the prior similar analyses.

We would like to thank the reviewers for suggesting to update our review since we can now include edoxaban which provides novelty to our network meta-analysis. We have included a more fulsome discussion of previously published network meta-analyses on this topic in the discussion section. We have also included a Table in Appendix 13 which summarizes the findings of our network meta-analyses in relation to other key published network meta-analyses published on the topic since publication of the article by Harenberg et al.¹

"To the best of our knowledge, this is the most up-to-date systematic review and network meta-analysis^{1–5} to synthesize the available benefits and harms on newer oral anticoagulants, standard adjusted dose VKA, and antiplatelet agents for the prevention of stroke and major bleeding in patients with non-valvular AF (Appendix 13). Harenberg et al¹ recently summarized the published indirect comparisons and network meta-analyses evaluating newer oral anticoagulants for AF and an earlier unpublished version of our analysis⁶ was considered among the most comprehensive analyses at that time.¹ This network meta-analysis expands upon that unpublished report.⁶ Overall, our findings align with those reported in previously published network meta-analyses;^{1,2,4} however, unlike previous analyses,¹ we include recently published edoxaban data,⁷ and also include comparisons with antiplatelet agents. We also present detailed evidence networks⁸ to illustrate the body of evidence for each outcome; report analysis using multiple statistical models (Appendix 9); report detailed a priori sub-group analyses for each outcome using data derived from FDA Public Summary Documents;^{9,10} present findings on both the relative and absolute scale,^{11,12} and graphically illustrate results using icon arrays. Our network meta-analysis is also the only one published to date that was registered in PROSPERO, and also adheres to PRISMA reporting standards (Appendix 10). It also differs from a recently published traditional meta-analysis by Ruff et al³ given we report comparison of newer oral anticoagulants versus each other, and also include comparisons with antiplatelet agents."

2. The authors indicate that the search for publications to include in the paper went through June 8, 2012, which is now nearly 1.5 years ago. The search for new trials to include in this analysis must be updated.

We have updated the search and re-ran the network meta-analysis including all relevant studies published until January 23, 2014.

3. Why did the authors not include larger phase II trials with NOACs, some of which randomized more than 1000 patients?

We have added Appendix 14 which provides rationale for why each study was excluded from the analysis, including the phase II trials with larger sample sizes. A systematic bias against phase II trials did not exist, rather, inclusion/exclusion of studies was driven by our published systematic review protocol (PROSPERO registry - CRD42012002721). We did include two phase II studies (PETRO and ARISTOTLE-J). However, the remaining phase II NOAC in AF trials had one or more factors that led to their exclusion from our meta-analysis (based on our predefined protocol), including (but not limited to): too small a sample size; too short a duration; inappropriate doses of NOAC; inappropriate treatment dose; use of an inappropriate (e.g. too wide) INR range for warfarin arm(s); variable definitions of primary outcome(s) outcomes; and insufficient information on key patient characteristics (e.g. CHADS score). Studies could also be excluded if the relevant information required by the systematic review protocol could not be identified.

Nonetheless, even had we applied less rigorous inclusion/exclusion criteria and included more phase II studies, the overall findings and conclusion of our analysis would not change significantly, especially for the newer oral anticoagulants. The phase II studies (even those with 1000 patients) would have contributed negligibly towards effect estimates given the substantially larger sample sizes of included Phase III studies. For example, another meta-analysis by Dentali et al¹³ applied less stringent inclusion criteria than our analysis, and larger phase II trials such as J-ROCKET contributed 0.3% of weight towards the effect estimate.

4. Since data with the largest RCT of a NOAC vs warfarin will be available in November 2013, the authors should include these latest data with edoxaban.

The data for edoxaban was not available when we submitted the manuscript. The revised manuscript now includes edoxaban data from ENGAGE AF-TIMI 487 in the reference case analysis.

Unfortunately, the ENGAGE AF-TIMI 487 publication does not provide raw data for sub-groups (i.e., number of patients and number of patients who had an event for each subgroup) and we were unable to report detailed sub-group analyses findings for edoxaban in Table 1. The data is also not available on Clinicaltrials.gov. We considered estimating the numbers using event rate data in the Supplementary Appendix but opted not to due to the potential risk of error. We would be willing to include this information in Table 1 if the investigators of ENGAGE AF-TIMI 48 trial⁷ would provide the data. It is our understanding that Reviewer #1 is an author on the paper. Otherwise, we have inserted a footnote highlighting that this data is currently not available for edoxaban.

5. How did the authors handle data where there were 2 experimental arms compared to the same control (e.g., RE-LY). What methods were used in the modeling and statistical analyses to avoid extra weight assigned to a control group (e.g., warfarin in RE-LY) that served as the same comparator group for 2 of the NOAC therapies (e.g., dabigatran 150 BID and dabigatran 110 BID in RE-LY)?

We used a binomial likelihood model which allows for the use of multi-arm trials and between-arm correlations between parameters are taken into account.^{14,15} We have added a sentence to the methods section which clearly indicates this: "A binomial likelihood model¹⁵ which accounts for the use of multi-arm trials was used for the analyses given the datasets provided were dichotomous outcomes and included multi-arm trials."

6. The benefit-harm assessment is an imbalanced one, as the authors have selected "major" bleeding for the harm vs. stroke/SEE for benefit. While some types of "major" bleeding carry a poor prognosis that is similar in severity to a stroke or SEE, others (such as a 2 gm/dL decline in hemoglobin) do not. The authors should repeat this analysis striking a better balance (e.g., use life-threatening bleeding or ICH as the harm vs. ischemic stroke/SEE as the benefit) or assign weighted utilities to these events. Otherwise, this is not really a fair balance of benefit and harm. This becomes a quite obvious limitation when inspecting Figure 4 – placebo/no treatment finishes a close 2nd to apixaban with only 1 more event per thousand (summing stroke + SEE + major bleed).

We agree with the review that both "major bleeding episodes" and "strokes /SEE" are heterogeneous diseases with broad ranges of clinical consequences. Admittedly, one of the definition criteria for major bleeding episode includes: "overt bleeding" with a decrease in the hemoglobin of 2g/dL. Although it may appear to not carry a poor prognosis, it is important to recognize that using this definition of major bleeding, a significant proportion (9 to 13%) of all major bleeding episodes will be fatal.¹⁶ Similarly, not all stroke or SEE will lead to significant morbidity and mortality. The most recently published study assessing edoxaban reported a rate of death or disabling stroke of approximately 11 to 13% per arm.⁷ Furthermore, the extent of disability from the stroke/SEE has not

been reported within the included studies. Therefore, we felt that using the primary efficacy and safety outcome measures of the included studies was the most reasonable option for our benefit-harm assessment. Furthermore, the net clinical benefit assessment of the ARISTOTLE trial¹⁷ also used a combination of major bleeding and “stroke/SEE” emphasizing the fact that it is a reasonable analysis. Similarly, the ENGAGE AF-TIMI 487 used “major bleeding” and “stroke/SEE” as the main outcomes for comparing efficacy and safety. Nonetheless, in order to account for the reviewer’s comment, we have modified the limitation Section of the manuscript to put more emphasis on this issue. “We report our findings using icon arrays, although it should be noted that these results do not account for the utility values based upon patient preferences for each of the outcomes,^{18,19} nor do they reflect uncertainty although we report absolute reductions on the risk benefit plane reflecting uncertainty in Appendix 8. There are other methods available that can incorporate patient preferences for outcomes.^{18,19} Further, estimates of benefit and harm with several of the therapies (e.g., rivaroxaban) come from one trial and thus such data are not particularly robust. Further research is needed comparing the balance of benefit and harms using other research methodologies, incorporating other relevant outcomes, patient preferences for each of the outcomes,^{18,19} and additional studies when they become available.”

We also acknowledge that the reviewer raises a very interesting point regarding placebo/no treatment which we discuss in more detail when addressing Comment #7 below.

7. A casual review of Figure 2 might lead the reader to believe that “no treatment/placebo” is not such a bad option, since the confidence bound for benefit overlaps 1.0 (not worse than VKA), and on the safety side, no treatment/placebo tends to look better than VKA therapy. Yet, this differs from the conclusions of prior VKA vs placebo meta-analyses that are the cornerstone of the guideline recommendations. What explains the difference and how do the authors propose to respond to the argument that no treatment/placebo isn’t an equally good option as compared to VKA?

We thank the reviewer for raising this important issue. We agree with the reviewer that previous systematic review and meta-analyses have reported a significant relative risk reduction in stroke or SE for patients with atrial fibrillation receiving warfarin compared to placebo.²⁰ Unfortunately, older reviews²⁰ often had broader inclusion criteria and included many studies that did not meet our pre-defined inclusion criteria. The protocol was published online and was registered in the PROSPERO International Prospective Register of Systematic Reviews in July 2012 (PROSPERO registry - CRD42012002721).

It is important to note that our reported point estimate is similar to those reported in previously published meta-analyses²⁰ but, as noted by the reviewer, the lower bound of the 95% confidence interval is crossing 1.0, likely because we included fewer studies due to our strict inclusion criteria, with only one study comparing warfarin with placebo directly. Had we expanded our network to include placebo comparisons with other warfarin doses (i.e., not restricted to a target international normalised ratio (INR) 2.0-3.0) this would likely improve precision for the placebo estimate; however, this would go against our pre-defined inclusion criteria. Further, this non-statistically significant finding is also partially explained by the differences in patient and study characteristics in the studies driving the estimate for placebo/no treatment. Two of these three studies making up the estimate compared placebo with aspirin (versus warfarin) and focused on “lower risk” patients, whereas only one study compared placebo with warfarin who would be at a higher risk. In contrast, the other treatments in the network are often always compared with warfarin and comprised of patients who would not be considered a low-risk of stroke/SE. Collectively, these help to explain why our findings for placebo versus warfarin differ from those reported in previous meta-analyses²⁰ which are the cornerstone of the guideline recommendations.

In retrospect, we could have excluded the placebo/no treatment node from the figures given the

issues noted above. Placebo/no treatment is really no longer a treatment option in the management of atrial fibrillation. This is echoed when examining the dates of studies which included placebo/no treatment where all placebo/no treatment studies included in our analysis were published at least 8 years ago, with the majority published more than 20 years ago. As such, these findings are likely not applicable to clinical practice today where placebo/no treatment is unlikely to be considered a reasonable treatment option in light of the evolution of clinical evidence. Based on these reasons, we have only included treatment options in the main text/figures (i.e., removed the placebo/no treatment), but included placebo/no treatment in the analysis to harness more information (and enhance precision and link other treatments) on the efficacy of aspirin. For completeness, however, we have included findings for no treatment/placebo in Appendix 6.

8. In Figure 4, how did the authors account for the fact that haemorrhage stroke is also a major bleed? Were these events double-counted? If not, to which group were they ascribed?

Figure 4 is based on the stroke/SE and major bleeding, the primary efficacy and safety outcomes for the majority of studies included in the analysis. We focused on these outcomes because they typically report sufficient data to permit sub-group analyses. Also, other recent meta-analyses and network meta-analyses have taken a similar approach. However, as the reviewer correctly notes, haemorrhage stroke is often counted in both the Stroke/SEE and major bleeding outcomes in the underlying studies. Unfortunately, there is insufficient data for some studies, especially the older studies with aspirin to subtract the haemorrhage stroke from the major bleed (or Stroke or SEE) to consistently account for this limitation. Further, this data is often not reported available for sub-group analyses. Rather than unfairly subtract it for some studies and sub-groups and not others, we decided to be consistent in our approach. It is important to note that this inherent double counting of haemorrhagic stroke may bias results in favor of the newer oral anticoagulants where they were consistently associated with lower numbers compared to warfarin. We have noted this limitation in the discussion section and added a footnote in Figure 4.

Minor issues (in order of appearance)

1. The wording in the abstract results are a bit confusing and hard to follow with regard to which therapy(ies) are being compared to which. Greater clarity is necessary here so that the review can easily understand the various comparisons being made.

We have explicitly included the comparator for each result.

2. Page 1, 2nd paragraph of the introduction. This is all well-described in the literature that includes dozens of meta-analyses (and several network analyses) on this topic. The paragraph could be substantially shortened.

We have shortened this paragraph.

3. Page 3, methods: Please define high, medium, and low dose ranges for aspirin.

We have included a sentence describing high, medium, and low dose ranges for aspirin: "Any dose of ASA was considered for inclusion, but ASA dose was stratified in the analysis as low (≤ 100 mg daily), medium (>100 mg to ≤ 300 mg daily), or high (>300 mg daily)."

4. Page 3, methods, last paragraph: Unclear if the authors excluded all the data from a study because of one data element with a zero cell (which would be undesirable), or just did not include a trial with a zero cell from that specific analysis.

We only excluded studies for analyses if all arms of the study had zero cells. We also excluded

treatments (e.g., apixaban 2.5 mg) which did not report at least one patient who had an event. We have included a sentence in the methods section indicating this: "Trials with zero cells in both arms or nodes where there were no events were excluded from the evidence networks because they do not contribute information or allow interpretable information.¹⁵ "

5. Page 4, ROCKET-AF included data stratified by CHADS2 score. The RE-LY are no longer the most recent.

ROCKET-AF only included patients with CHADS2 score greater than or equal to 2. We have added a statement in the methods section clearly indicating why the RE-LY trial was chosen: "The RE-LY trial was selected because it contained data for both CHADS2<2 and CHADS2≥2 and had detailed data available from the Food and Drug Administration (FDA) Public Summary Report. We also indicate in the discussion section that this assumption likely has limited impact on results: "Use of a different study (e.g., ARISTOTLE) for baseline event rate data is unlikely to change the results substantially given that the event rates in standard adjusted dose VKA arms were similar across studies."

6. The authors should register this analysis online (e.g., PROSPERO or metcardio.org?)

This analysis was registered in PROSPERO in July 2012 (PROSPERO registry - CRD42012002721).

7. The authors should acknowledge that the estimates of benefit and harm with several of the therapies (e.g., rivaroxaban) come from 1 trial, and thus such data are not particularly robust.

We have added this to the limitations section: "We report our findings using icon arrays, although it should be noted that these results do not account for the utility values based upon patient preferences for each of the outcomes, nor do they reflect uncertainty although we report absolute reductions on the risk benefit plane reflecting uncertainty in Appendix 8. There are other methods available that can incorporate patient preferences for outcomes. Further, estimates of benefit and harm with several of the therapies (e.g., rivaroxaban) come from one trial and thus such data are not particularly robust. Further research is needed comparing the balance of benefit and harms using other research methodologies, incorporating other relevant outcomes, and patient preferences for each of the outcomes, and additional studies when they become available."

8. Figure 1, some of the nodes are either too small or missing (e.g., clopidogrel + aspirin). If too small, then size up all the nodes so that it is possible to see even the node that corresponds to the treatment with the least number of patients. This comment also applies to Appendices 5, 12A, 12B.

We have revised all network figures so all the nodes are visible.

9. Figure 2: the size of the point estimate should be proportional to the number of patients (or trials) studied with that therapy.

The size of the study is reflected within the standard error which is captured in the Credible Intervals in Figure 2. This is a more methodologically appropriate than adjusting by sample size alone.

10. Figure 4 – I like this figure, but suggest separate rows for the blue and red avatars. That would facilitate a quick comparison of each element (stroke/SEE, major bleeding). It would also dissuade the reader from simply comparing the sum of red+blue across treatments. If one did do that, placebo is 2nd best only to apixaban (and differ by only 1 patient per thousand)! See major comment #7.

Unfortunately, the software package (www.iconarray.com) does not permit us to use separate rows for individual outcomes. We are working on developing software for allowing this in the future.

We have also addressed the comment on placebo above – See response to comment #7.

11. Appendix 8 is an interesting plot. It could be improved by selecting a type of bleeding that is similar in severity to a stroke/SEE, or by changing the scale of the Y-axis to account for a difference in weighting between major bleeding and stroke/SEE.

See response to comment #6.

Reviewer 2 (Andreas Goette):

Unfortunately, the ENGAGE AF study was presented during the AHA 2013 in Dallas. The study included more than 21.000 patients. The general term of the presented manuscript is therefore misleading. The authors must include the data from the ENGAGE AF study. In addition, they must show how their results are related to the latest meta-analysis encompassing all NOAC's (see Ruff et al The Lancet 2013)

The ENGAGE AF-TIMI 48 study⁷ not available when we submitted the manuscript. We have updated the search and re-ran all analyses including all relevant studies published until January 23, 2014, including the ENGAGE AF-TIMI 48 study.⁷ We have also included a sentence discussing how our analysis differs from the publication by Ruff et al³ in Lancet 2013: It also differs from a recently published traditional meta-analysis by Ruff et al³ given we report comparison of newer oral anticoagulants versus each other, and also include comparisons with antiplatelet agents. We also added Appendix 13 which discussed our analysis in comparison with other published network meta-analyses: "To the best of our knowledge, this is the most up-to-date systematic review and network meta-analysis^{1–5} to synthesize the available benefits and harms on newer oral anticoagulants, standard adjusted dose VKA, and antiplatelet agents for the prevention of stroke and major bleeding in patients with non-valvular AF (Appendix 13). Harenberg et al¹ recently summarized the published indirect comparisons and network meta-analyses evaluating newer oral anticoagulants for AF and an earlier unpublished version of our analysis⁶ was considered among the most comprehensive analyses at that time.¹ This network meta-analysis expands upon that unpublished report.⁶ Overall, our findings align with those reported in previously published network meta-analyses;^{1,2,4} however, unlike previous analyses,¹ we include recently published edoxaban data,⁷ and also include antiplatelet agents. We also present detailed evidence networks⁸ to illustrate the body of evidence for each outcome; report analysis using multiple statistical models (Appendix 9); report detailed a priori sub-group analyses for each outcome using data derived from FDA Public Summary Documents;^{9,10} present findings on both the relative and absolute scale,^{11,12} and graphically illustrate results using icon arrays. Our network meta-analysis is also the only one published to date that was registered in PROSPERO, and also adheres to PRISMA reporting standards (Appendix 10). It also differs from a recently published traditional meta-analysis by Ruff et al³ given we report comparison of newer oral anticoagulants versus each other, and also include comparisons with antiplatelet agents."

Reviewer 3 (Prof J L Hutton):

This is a very thorough, careful study and report.

Thank you for your comments

Minor points:

Page 15, line 22: 'with in' - should this be 'with increasing'? Something is missing.

We have made the suggested change

Page 17, line 15 'discussion' not 'discussing'

We have made the suggested change

Page 29, ref 34 - The two lower case 'a's are presumably wrong? The first should be 'A', the second I'm not sure.

We have made the suggested change

I have one concern: the completion rates for the studies varied from 72% to 92%. The largest study had on 74% completion. Some assessment of whether there was differential completion by drug, and the sub-group factors would be sensible, with a brief comment.

We have added a sentence in the discussion section indicating that completion rates varied among treatments; however, given the connections in the network were comprised of single study connections we had limited ability to adjust for this issue in our analysis. We have also added a box-plot in Appendix 15 to show this graphically. "First, there is notable heterogeneity in patients and study characteristics (Appendix 3, 4, and 16). For example, there were differences in patients enrolled in the underlying studies and completion rates. However, the small number of studies limits the analyses that can be conducted to account for heterogeneity in the absence of patient-level data."

I don't think further review is required.

In summary, we again thank the editor and reviewers for their helpful comments and suggestions. We hope we have addressed them successfully and the manuscript has been revised accordingly.

We thank you for considering our revised manuscript for publication in your journal and look forward to hearing from you in the near future.

Sincerely,
Chris Cameron

References (for responses)

1. Harenberg J, Weiss C. Clinical trials with new oral anticoagulants. Additive value of indirect comparisons also named network meta-analyses. *Hamostaseologie*. 2013;33(1):62–70.
2. Dogliotti A, Paolasso E, Giugliano RP. Current and new oral antithrombotics in non-valvular atrial fibrillation: a network meta-analysis of 79 808 patients. *Heart*. 2013.
3. Ruff CT, Giugliano RP, Braunwald E, et al. Comparison of the efficacy and safety of new oral anticoagulants with warfarin in patients with atrial fibrillation : a meta-analysis of randomised trials. *Lancet*. 2013;6736(13):1–8.
4. Assiri A, Al-Majzoub O, Kanaan AO, Donovan JL, Silva M. Mixed treatment comparison meta-analysis of aspirin, warfarin, and new anticoagulants for stroke prevention in patients with nonvalvular atrial fibrillation. *Clin. Ther*. 2013;35(7):967–984.e2.
5. Kmh BS, Berge E. Factor Xa inhibitors versus vitamin K antagonists for preventing cerebral or systemic embolism in patients with atrial fibrillation (Review). 2013;(8).
6. Wells G, Coyle D, Cameron C, Steiner S, Kelly S. Safety, effectiveness, and costeffectiveness of new oral anticoagulants compared with warfarin in preventing stroke and other cardiovascular events in patients with atrial fibrillation.; 2012. Available at: http://www.cadth.ca/media/pdf/NOAC_Therapeutic_Review_final_report.pdf.

7. Giugliano RP, Ruff CT, Braunwald E, et al. Edoxaban versus warfarin in patients with atrial fibrillation. *N. Engl. J. Med.* 2013;369(22):2093–104.
8. Salanti, Georgia, Kavvoura, F, Ioannidis J. Exploring the geometry of treatment networks. *Ann. Intern. Med.* 2008;148:544–553.
9. Ingelheim B. Advisory committee briefing document: dabigatran etexilate [Internet].; 2010:1–168. Available at:
<http://www.fda.gov/downloads/advisorycommittees/committeesmeetingmaterials/drugs/cardiovas>.
10. Johnson & Johnson Pharmaceutical Research & Development. Rivaroxaban for the Prevention of Stroke and Non-Central Nervous System (CNS) Systemic Embolism in Patients With Atrial Fibrillation.; 2011:222. Available at:
<http://www.fda.gov/downloads/AdvisoryCommittees/CommitteesMeetingMaterials/drugs/CardiovascularandRenalDrugsAdvisoryCommittee/ucm270797.pdf>.
11. Barratt A, Wyer PC, Hatala R, et al. Tips for learners of evidence-based medicine: 1. Relative risk reduction, absolute risk reduction and number needed to treat. *Can. Med. Assoc. J.* 2004;171(4):353–8.
12. Schechtman E. Odds ratio, relative risk, absolute risk reduction, and the number needed to treat--which of these should we use? *Value Health.* 2002;5(5):431–6.
13. Dentali F, Riva N, Crowther M, Turpie AGG, Lip GYH, Ageno W. Efficacy and safety of the novel oral anticoagulants in atrial fibrillation: a systematic review and meta-analysis of the literature. *Circulation.* 2012;126(20):2381–91.
14. Dias S, Sutton AJ, Ades a E, Welton NJ. Evidence synthesis for decision making 2: a generalized linear modeling framework for pairwise and network meta-analysis of randomized controlled trials. *Med. Decis. Making.* 2013;33(5):607–17.
15. Dias S, Welton NJ, Sutton AJ, Ades A. NICE DSU Technical Support Document 2: A Generalised Linear Modelling Framework for Pairwise and Network Meta-Analysis of Randomised Controlled Trials.; 2011:1–96.
16. Majeed A, Hwang H-G, Connolly SJ, et al. Management and outcomes of major bleeding during treatment with dabigatran or warfarin. *Circulation.* 2013;128(21):2325–32. Available at:
<http://www.ncbi.nlm.nih.gov/pubmed/24081972>. Accessed February 4, 2014.
17. Granger C, Alexander J. Apixaban versus warfarin in patients with atrial fibrillation. *N. Engl. J. Med.* 2011:981–992.
18. Guo JJ, Pandey S, Doyle J, Bian B, Lis Y, Raisch DW. A review of quantitative risk-benefit methodologies for assessing drug safety and efficacy-report of the ISPOR risk-benefit management working group. *Value Health.* 2010;13(5):657–66.
19. Boyd C, Singh S, Varadhan R, et al. Methods Research Report Methods for Benefit and Harm Assessment in Systematic Reviews. Rockville, MD; 2012:52.
20. Hart R, Pearce L, Aguilar M. Meta-analysis: antithrombotic therapy to prevent stroke in patients who have nonvalvular atrial fibrillation. *Ann. Intern. Med.* 2007.

VERSION 2 – REVIEW

REVIEWER	Robert Giugliano Brigham and Women's Hospital Harvard Medical School Boston, MA USA I am an investigator in the ENGAGE AF-TIMI 48 Trial sponsored by Daiichi-Sankyo, and have received honoraria for consulting or CME lectures from Bristol Myers Squibb, Daiichi-Sankyo, Johnson & Johnson, and Pfizer.
REVIEW RETURNED	07-Mar-2014

GENERAL COMMENTS

The authors are to be congratulated on this comprehensive and very detailed network meta-analysis that extends beyond prior similar publications. Their methods are sound, data are robust, and conclusions appropriate, excepting a few issues noted below. That being said, I do have several comments / suggestions to improve the accuracy and reporting of the data that pertain to 1) The authors' responses to the initial comments from the referees, 2) New data/errors introduced since the initial submission, 3) Additional minor comments.

Author responses to initial review.

1. The authors should temper the statement that this manuscript is the first or only network meta-analysis registered on PROSPERO (page 11, line 52), as I am aware of at least one other.
2. It would be most helpful to the reader to include the full entry criteria for which studies were included vs excluded should appear in the methods section of the paper.
3. Why wasn't the trial by Weitz et al (Thromb Haemost 2010;104:633-41), a phase II study of edoxaban, included in this meta-analysis? This is not clear from the entry criteria described in the methods.
4. I would encourage the authors to request data that are currently missing in Table 1 from specific trials by writing to the responsible investigator / research group.
5. I remain very concerned about showing only a single net outcome that includes endpoints of very different weight (i.e., major bleeding is frequent and generally with less consequence than stroke or death). Can the authors perform a sensitivity analysis that includes a higher level of bleeding (e.g., ICH or life-threatening bleeding).
6. In some countries (e.g., India) oral anticoagulation is not widely used in patients with AF who are not at very high risk. As such, showing data that suggest no treatment/placebo may be acceptable might be misconstrued as evidence in support of a "no treatment" option. The findings with "no treatment", if shown, should be put into better context.
7. I encourage the authors to generate a version of Figure 4 that better separates the blue and red avatars, recognizing limitations in the software currently available.

Critical comments

1. Page 15, line 39. The authors mean "decreased" not "increased".
2. Figure 2B - the confidence intervals indicated numerically for several of the treatments have changed dramatically and do not match the figure. For example apixaban 5 mg used to 0.69 (0.6, 0.8), but is now 0.68 (0.25, 1.35). Yet the graphic shows negligible change from the prior version and neither approaches anywhere near 0.25 and does not cross 1.0. This same mismatch is true of most of the elements in Figure 2B. Which is correct the figure or the numeric data? If the latter, then this represents a huge change from the initial submission and deserves explanation.

Other minor comments

1. Page 16, line 41. The authors note that some data are "reported elsewhere". This is vague and confusing. Please either provide the citation or state that the "data are not shown."
2. Appendix 4 - please explain how you defined "Completed", i.e., what factors could lead to a patient being categorized as incomplete. Also, such a statistic is potential misleading if it does not account for the highly variable length of follow-up seen in this dataset. One would expect in comparing two trials of similar quality/population that

	the longer trial would have a higher rate of non-completers. 3. Appendix 7 - showing that No treatment/placebo has a 70.6% chance of probability of being ranked the best with regard to bleeding is not helpful, since we know that antithrombotic agents can only increase bleeding. 4. Appendix 8 - The color choices are confusing here. Better to keep the two doses of dabigatran two shades/patterns of red, and the two doses of edoxaban two shades/patterns of green, to avoid confusion. 5. Appendix 9 Forest plot using vague priors -- The results of the Random Effects Model with Vague Prior yields extremely broad CIs and are not helpful to the reader. For example, no reasonable person would believe an upper bound for dabigatran-150 of 4.79 for prevention of stroke/SEE. 6. Appendix 15 is not necessary. It does not add to what is already presented in the table.
--	---

VERSION 2 – AUTHOR RESPONSE

Reviewer Name: Robert Giugliano

The authors are to be congratulated on this comprehensive and very detailed network meta-analysis that extends beyond prior similar publications. Their methods are sound, data are robust, and conclusions appropriate, excepting a few issues noted below.

Thank you for the kind words.

That being said, I do have several comments / suggestions to improve the accuracy and reporting of the data that pertain to 1) The authors' responses to the initial comments from the referees, 2) New data/errors introduced since the initial submission, 3) Additional minor comments.

Author responses to initial review.

1. The authors should temper the statement that this manuscript is the first or only network meta-analysis registered on PROSPERO (page 11, line 52), as I am aware of at least one other.

We have tempered our statement. We now only state that “our review was registered in PROSPERO”

2. It would be most helpful to the reader to include the full entry criteria for which studies were included vs excluded should appear in the methods section of the paper.

We have included the full entry criteria for which studies were included vs excluded in the main text of the methods: “Active and placebo-controlled randomized controlled trials (RCTs) were selected for inclusion if they were published in English, included at least one antithrombotic treatment under review (using pre-specified doses), reported data for any of the pre-specified outcomes related to patient stroke/SE or major bleeding, and involved patients with non-valvular AF eligible to receive anticoagulant therapy, regardless the level of stroke risk. Trials that included patients with contraindication to anticoagulant treatment were excluded. VKA trials were included if the dose was adjusted to a target international normalised ratio (INR) 2.0-3.0. Any dose of ASA was considered for inclusion, but ASA dose was stratified in the analysis as low (≤ 100 mg daily), medium (>100 mg to ≤ 300 mg daily), or high (>300 mg daily). We only included new oral anticoagulants which had at least one large phase III study.”

3. Why wasn't the trial by Weitz et al (Thromb Haemost 2010;104:633-41), a phase II study of edoxaban, included in this meta-analysis? This is not clear from the entry criteria described in the methods.

Edoxaban was not included in the original systematic review that was recently updated. The search strategy was updated to identify new studies published since our original review and as a result Phase II studies published on edoxaban before the original systematic review search date such as the study by Weitz et al(1) were not captured. This did not impact other treatments given they were all included in the original review. We have revised the analysis to include the data for once daily doses of edoxaban (doses used in ENGAGE AF-TIMI 48 trial(2)) from the Weitz et al trial.(1) We also identified other Phase II edoxaban studies may have been missed due to the issue noted above by checking ClinicalTrials.gov, PubMed, and reviewing reference lists in other published systematic reviews and meta-analyses.(3) We identified two other Phase II studies(4,5) but these studies only reported data for major bleeding. As expected, the results for Stroke/SE did not change after including Weitz et al(1) trial given the small sample size of the Weitz et al1 in comparison to the ENGAGE AF-TIMI 48 trial.(2) The results for major bleeding also did not change substantially when Weitz et al1, Chung et al,4 and Yamashita et al5 were added given collectively they are small in comparison with the ENGAGE AF-TIMI 48 trial.(2)

4. I would encourage the authors to request data that are currently missing in Table 1 from specific trials by writing to the responsible investigator / research group.

We requested and obtained the data from the responsible investigator / research group and re-ran sub-group analyses including this data. These findings are now reported in Table 1.

5. I remain very concerned about showing only a single net outcome that includes endpoints of very different weight (i.e., major bleeding is frequent and generally with less consequence than stroke or death). Can the authors perform a sensitivity analysis that includes a higher level of bleeding (e.g., ICH or life-threatening bleeding).

We appreciate the reviewers concern and started conducting a sensitivity analysis that included a higher level of bleeding (e.g., ICH or life-threatening bleeding) but opted not to continue with the requested analysis for a number of reasons:

- Life-threatening bleeding data was not consistently reported among included studies.
- An analysis for ICH would be inappropriate given ICH is double counted in stroke/SE (as hemorrhagic strokes) and major bleeding within underlying studies (and data is not consistently available to subtract out). At least by including major bleeding (versus ICH or life-threatening bleeding) as the bleeding outcome, we are able to minimize the potential bias resulting from the double counting of ICH within the stroke/SE and major bleeding within underlying studies.
- The double counting issue would remain a bigger concern for life-threatening bleeding (in comparison with major bleeding) even if data was consistently reported among included studies
- The submitted manuscript is part of a broader research agenda comparing new oral anticoagulants for a variety of clinical areas, with atrial fibrillation being one of them. For consistency with other reviews on new oral anticoagulants (one published(6) and the other submitted for publication), we have focused on the primary efficacy and safety endpoints.
- By selecting the primary efficacy and safety outcome consistency among the various reviews, we cannot be criticized for selectively choosing particular outcomes to assess benefits and harms.
- Finally, being consistent in our approach among reviews will allow us (and others) to compare and contrast findings among the various reviews we produce on new oral anticoagulants in different clinical areas moving forward.

Nonetheless, we appreciate the issues noted by the reviewer and have discussed them in detail in the discussion section: "We report our findings using icon arrays, although it should be noted that these results do not account for the utility values based upon patient preferences for each of the outcomes,(7,8) nor do they reflect uncertainty although we report absolute reductions on the risk benefit plane reflecting uncertainty in Appendix 8. There are other methods available that can incorporate patient preferences for outcomes.(7,8) Further, estimates of benefit and harm with several of the therapies (e.g., rivaroxaban) come from one trial and thus such data are not particularly robust. Further research is needed comparing the balance of benefit and harms using other research methodologies, incorporating other relevant outcomes, patient preferences for each of the outcomes,(7,8) and additional studies when they become available. Finally, haemorrhagic stroke is also considered a major bleed in underlying studies. Accordingly, we may have double counted haemorrhage stroke in this analysis. Unfortunately, we are not able to account for this issue in all included studies. Nonetheless, it is important to note this limitation and highlight that this potentially biases results in favor of newer oral anticoagulants given these agents were associated with reductions in these events."

6. In some countries (e.g., India) oral anticoagulation is not widely used in patients with AF who are not at very high risk. As such, showing data that suggest no treatment/placebo may be acceptable might be misconstrued as evidence in support of a "no treatment" option. The findings with "no treatment", if shown, should be put into better context.

We have not reported findings from the "no treatment option" in the main text to help prevent this issue. The data for no treatment was only included to improve precision and findings for no treatment are only reported in the supplemental appendices for completeness.

7. I encourage the authors to generate a version of Figure 4 that better separates the blue and red avatars, recognizing limitations in the software currently available.

We appreciate the reviewers' opinion on these figures. However, as noted in comment 5 this review is part of a broader research project comparing new oral anticoagulants for a variety of clinical areas, with atrial fibrillation being just one of them. To align the icon arrays reported in other reviews on new oral anticoagulants (one published(6) and the other submitted for publication), we have focused on the primary efficacy and safety endpoint in each analysis. For other publications,(6) we have not separated the blue and red avatars for the primary safety and efficacy outcomes given icon arrays do not typically separate avatars and currently available software does not allow one to easily do so (<http://www.iconarray.com/>). As such, we would prefer to keep the icon arrays in the similar format to aid in comparison among the different tables/figures reported among the various reviews on new oral anticoagulants, although we appreciate the reviewers preference.

Critical comments

1. Page 15, line 39. The authors mean "decreased" not "increased".

We have changed the text accordingly

2. Figure 2B - the confidence intervals indicated numerically for several of the treatments have changed dramatically and do not match the figure. For example apixaban 5 mg used to 0.69 (0.6, 0.8), but is now 0.68 (0.25, 1.35). Yet the graphic shows negligible change from the prior version and neither approaches anywhere near 0.25 and does not cross 1.0. This same mismatch is true of most of the elements in Figure 2B. Which is correct the figure or the numeric data? If the latter, then this represents a huge change from the initial submission and deserves explanation.

Thank you for noticing this error. The findings for the random-effects model were reported in Figure 2B. We have updated to report the results for the fixed-effects model to align with the other sections of the manuscript. We provide the rationale for choice of fixed-effects model in Appendix 9.

Other minor comments

1. Page 16, line 41. The authors note that some data are "reported elsewhere". This is vague and confusing. Please either provide the citation or state that the "data are not shown."

We have included the reference and deleted this statement.

2. Appendix 4 - please explain how you defined "Completed", i.e., what factors could lead to a patient being categorized as incomplete. Also, such a statistic is potential misleading if it does not account for the highly variable length of follow-up seen in this dataset. One would expect in comparing two trials of similar quality/population that the longer trial would have a higher rate of non-completers.

We defined "completed" as the number of participants at the end of the study period as the number of participants who completed the study, i.e. the number of participants who were not reported as having not completed the study for any reason (including lost to follow-up, non-compliance, AE, death, etc.). While the "completed" statistic is not universally reported, nor is there a universal definition, it is interesting to note that this is one of the two milestones for which Clinicaltrials.gov requires that data are reported, the other milestone being "started" – see http://prsinfo.clinicaltrials.gov/results_definitions.html#Result_ParticipantFlow.

As the number of participants who do not complete a study increases with the duration of a trial, the reviewer is correct in pointing out that for two trials that are identical except in duration, the longer trial would have a lower proportion of participants who are reported as "completed". Therefore, we appreciate that this statistic can be somewhat counterintuitive and potentially misleading. As this statistic is not central to any of the analysis or reporting in our manuscript, we have removed it from Appendix 4.

3. Appendix 7 - showing that No treatment/placebo has a 70.6% chance of probability of being ranked the best with regard to bleeding is not helpful, since we know that antithrombotic agents can only increase bleeding.

A consequence of including placebo in the analysis to improve precision is that it is included in the calculations of probability best. We recognized this potential issue from the outset and also included mean rank and the surface under the cumulative ranking (SUCRA) whenever we present probability best. The SUCRA accounts for the cumulative probability among multiple rankings. It would be 100% when a treatment is certain to be the best and 0% when a treatment is certain to be the worst. SUCRA is considered by some a better probability measure than probability best.⁽⁹⁾ We have added some discussion which discusses SUCRA to Appendix 7.

4. Appendix 8 - The color choices are confusing here. Better to keep the two doses of dabigatran two shades/patterns of red, and the two doses of edoxaban two shades/patterns of green, to avoid confusion.

We have changed the color scheme to align with the Reviewer's suggestion.

5. Appendix 9 Forest plot using vague priors -- The results of the Random Effects Model with Vague Prior yields extremely broad CIs and are not helpful to the reader. For example, no reasonable person would believe an upper bound for dabigatran-150 of 4.79 for prevention of stroke/SEE.

We agree there are issues with using the random-effects model with vague priors and have provided

a detailed discussion on our choice of statistical model within this appendix. However, best practices for the conduct of meta-analysis and network meta-analysis suggest that multiple statistical models should be considered and findings presented for each and random-effects analyses using vague priors is commonplace. For these reasons, we included the random-effects analysis using a vague prior. Nonetheless, we agree with reviewer that findings from a random-effects model using a vague prior are unrealistic and have inserted a statement clearly indicating this is Appendix 9: "The use of a random-effects model with a vague prior on the between study variance exerts a large degree of influence on the credible intervals (CrI's) because there is insufficient studies to reign in the prior and provide an accurate estimate of the between study variance. As a result, results often appear unrealistic (e.g., wide credible intervals for a treatment that was shown to be associated with a statistically significant decrease in stroke/SE). Accordingly, we also considered informative priors on the between study variance from the recently published reports.(12,44.)

We have also reordered Appendix 9 to show the findings from the random-effects model using informative priors first (i.e., before the random-effects model using vague priors) because these are more realistic.

6. Appendix 15 is not necessary. It does not add to what is already presented in the table.

We have removed appendix 15

In summary, we again thank the editor and reviewers for their helpful comments and suggestions. We hope we have addressed them successfully and the manuscript has been revised accordingly.

We have also acknowledged Dr. Robert Giugliano in the acknowledgements section of the manuscript for providing the sub-group data for edoxaban and also his helpful comments on the manuscript.

We again thank you for considering our revised manuscript for publication in your journal and look forward to hearing from you in the near future.

Sincerely,

Chris Cameron

References

1. Weitz JI, Connolly SJ, Patel I, et al. Randomised, parallel-group, multicentre, multinational phase 2 study comparing edoxaban, an oral factor Xa inhibitor, with warfarin for stroke prevention in patients with atrial fibrillation. *Thromb. Haemost.* 2010;104(3):633–41. Available at: <http://www.ncbi.nlm.nih.gov/pubmed/20694273>. Accessed February 16, 2014.
2. Giugliano RP, Ruff CT, Braunwald E, et al. Edoxaban versus warfarin in patients with atrial fibrillation. *N. Engl. J. Med.* 2013;369(22):2093–104. Available at: <http://www.ncbi.nlm.nih.gov/pubmed/24251359>. Accessed January 20, 2014.
3. Assiri A, Al-Majzoub O, Kanaan AO, Donovan JL, Silva M. Mixed treatment comparison meta-analysis of aspirin, warfarin, and new anticoagulants for stroke prevention in patients with nonvalvular atrial fibrillation. *Clin. Ther.* 2013;35(7):967–984.e2. Available at: <http://www.ncbi.nlm.nih.gov/pubmed/23870607>. Accessed February 7, 2014.
4. Chung N, Jeon H-K, Lien L-M, et al. Safety of edoxaban, an oral factor Xa inhibitor, in Asian patients with non-valvular atrial fibrillation. *Thromb. Haemost.* 2011;105(3):535–44. Available at: <http://www.ncbi.nlm.nih.gov/pubmed/21136011>. Accessed March 2, 2014.
5. Yamashita T, Koretsune Y, Yasaka M, et al. Randomized, Multicenter, Warfarin-Controlled Phase II Study of Edoxaban in Japanese Patients With Non-Valvular Atrial Fibrillation. *Circ. J.* 2012;76(8):1840–1847. Available at: <http://japanlinkcenter.org/DN/JST.JSTAGE/circj/CJ-11-1140?lang=en&from=CrossRef&type=abstract>. Accessed March 16, 2014.
6. Castellucci L, Cameron C. safety outcomes of oral anticoagulants and antiplatelet drugs in the secondary prevention of venous thromboembolism: systematic review and network meta-. *BMJ Br. Med.* 2013;5133(August):1–12. Available at: <http://www.ncbi.nlm.nih.gov/pmc/articles/PMC3758108/>. Accessed March 17, 2014.
7. Guo JJ, Pandey S, Doyle J, Bian B, Lis Y, Raisch DW. A review of quantitative risk-benefit methodologies for assessing drug safety and efficacy-report of the ISPOR risk-benefit management working group. *Value Health.* 2010;13(5):657–66. Available at: <http://www.ncbi.nlm.nih.gov/pubmed/20412543>. Accessed July 13, 2012.
8. Boyd C, Singh S, Varadhan R, et al. Methods Research Report Methods for Benefit and Harm Assessment in Systematic Reviews. Rockville, MD; 2012:52. Available at: www.ncbi.nlm.nih.gov/books/NBK115750/.
9. Salanti G, Ades A, Ioannidis J. Graphical methods and numerical summaries for presenting results from multiple-treatment meta-analysis: an overview and tutorial. *J. Clin. Epidemiol.* 2011;64(2):163–71. Available at: <http://www.ncbi.nlm.nih.gov/pubmed/20688472>. Accessed February 10, 2013.

VERSION 3 - REVIEW

REVIEWER	Robert Giugliano Brigham and Women's Hospital USA
REVIEW RETURNED	17-Apr-2014

GENERAL COMMENTS	 1. The data on edoxaban are now added, which adds substantially to the manuscript. However, these data have been entered into the bottom of tables, figures and the text. This doesn't make sense since the other new oral anticoagulants are at the top, then antiplatelets and then the data with edoxaban. Instead, the edoxaban data should appear with the other newer anticoagulants. 2. Response to Referee 1, AQ #5. The authors could have, but chose not to, presented data on stroke or ICH without double counting, by subtracting out the hemorrhagic strokes. This has been done by a number of other authors. Another alternative is to report subdural and epidural bleeds, which
--

	are ICHs, but not hemorrhagic strokes. 3. It is not true that the authors are above reproach (as they state in their response to AQ #5) because they opted to combine the primary efficacy and safety endpoints. These are clearly of different importance and clinical consequence. For example, the authors should compare the rates of mortality and major disability after stroke with that of a major bleed. Regulatory authorities have not accepted such composite endpoints that are imbalanced for good reason. Lastly, other authors have used various techniques (e.g., weighting or selecting only the more severe bleeding events) to construct a composite endpoint with more equitable components. 4. If the authors insist on combining events of unequal weight, this should be explicitly discussed as limitation #1 and their rationale explained. 1. Please move the data on edoxaban in the text (e.g., page 13), tables, and figures up so they are grouped with the other newer anticoagulants. Specifically, it makes more sense to discuss the safety results on page 13 starting with the drugs that have the least major bleeding and then proceeding with those that have more major bleeding. Right now, it looks like the data with edoxaban have just been appended at the end without consideration with where the data fit best. 2. Please reconsider reporting on stroke + ICH without double-counting, by subtracting out the hemorrhagic strokes 3. Please reconsider some sort of weighting of the composite of efficacy and safety. Alternatively, you could modify the safety endpoint to only include more severe major bleeding events, as this would be a more equitable composite of efficacy + safety (net outcome). Otherwise, this remains a major limitation of the analyses of net outcome that needs to be more explicitly discussed, with a rationale for why the analyses was performed in this way (in contrast to the approaches others have taken).
--	---

VERSION 3 – AUTHOR RESPONSE

Reviewer Name: Robert Giugliano

Institution and Country: Brigham and Women's Hospital USA

Please state any competing interests or state 'None declared': None

1. The data on edoxaban are now added, which adds substantially to the manuscript. However, these data have been entered into the bottom of tables, figures and the text. This doesn't make sense since the other new oral anticoagulants are at the top, then antiplatelets and then the data with edoxaban. Instead, the edoxaban data should appear with the other newer anticoagulants.

We have revised the manuscript to order treatments alphabetically in the methods sections of the report. We have also revised the results section of the manuscript ordering the reporting of findings in tables and text according to most pronounced benefit for that particular outcome, i.e., dabigatran 150mg twice daily discussed first for stroke/SE because it had the most pronounced benefit on stroke/SE, and edoxaban 30mg daily for major bleeding. For figures which report both stroke/SE and major bleeding within the same figure, we have ordered according to stroke/SE because this was the primary efficacy outcome for each study and has larger clinical consequences on patients in most

instances.

2. Response to Referee 1, AQ #5. The authors could have, but chose not to, presented data on stroke or ICH without double counting, by subtracting out the hemorrhagic strokes. This has been done by a number of other authors. Another alternative is to report subdural and epidural bleeds, which are ICHs, but not hemorrhagic strokes.

We are not comfortable changing our outcome definitions at this stage of the review for both methodological reasons, and due to concerns related to how a change this late in the research process may be perceived given the reviewer requesting the change is the primary author on one of the included studies – ENGAGE AF-TIMI 48. As discussed in our previous responses, we selected the primary efficacy and safety endpoint reported in underlying studies included in our review. Indeed, the reviewer requesting the revision used stroke/SE and major bleeding as the primary efficacy and safety outcomes in the study where he is listed as the primary. Also, as noted previously, this network meta-analysis is just one network meta-analysis that our group has undertaken and we have taken a similar approach in all analyses. While we appreciate the limitations noted by the reviewer, it will allow readers to compare findings with those reported in the underlying studies versus having discordant results due to subtracting outcomes from certain outcomes. Nonetheless, we have clearly acknowledged this limitation and the impact it likely has on results: “Finally, haemorrhagic stroke is also considered a major bleed in underlying studies. Accordingly, we may have double counted haemorrhage stroke in this analysis. Unfortunately, we are not able to account for this issue in all included studies. Nonetheless, it is important to note this limitation and highlight that this potentially biases results in favor of newer oral anticoagulants given these agents were associated with reductions in these events.”

3. It is not true that the authors are above reproach (as they state in their response to AQ #5) because they opted to combine the primary efficacy and safety endpoints. These are clearly of different importance and clinical consequence. For example, the authors should compare the rates of mortality and major disability after stroke with that of a major bleed. Regulatory authorities have not accepted such composite endpoints that are imbalanced for good reason. Lastly, other authors have used various techniques (e.g., weighting or selecting only the more severe bleeding events) to construct a composite endpoint with more equitable components.

See comment #2 where we indicate that we are not comfortable changing our outcomes at this stage of the review and refer to our previous responses to this comment in past drafts. Further, the reviewer implies that we are using a composite outcome in his response which is not the case. Our icon arrays provide both stroke/SE and major bleeding data in their disaggregated form (i.e., use different colors) which is not the case for composite outcomes where outcomes are lumped together. Further, at no point in our paper do we sum up the absolute risk results for stroke/SE and major bleeding which the reviewer is implying and is the case for composite outcomes. We simply report the findings from the underlying primary and safety outcomes reported in underlying studies visually in disaggregated fashion using icon arrays. We also clearly acknowledge this limitation: “We report our findings using icon arrays, although it should be noted that these results do not account for the utility values based upon patient preferences for each of the outcomes,^{42,43} nor do they reflect uncertainty although we report absolute reductions on the risk benefit plane reflecting uncertainty in Appendix 8. There are other methods available that can incorporate patient preferences for outcomes.^{42,43} Further, estimates of benefit and harm with several of the therapies (e.g., rivaroxaban) come from one trial and thus such data are not particularly robust. Further research is needed comparing the balance of benefit and harms using other research methodologies, incorporating other relevant outcomes, patient preferences for each of the outcomes,^{42,43} and additional studies when they become available.”

Further, we have revised the ordering of the icon arrays in the latest version of our manuscript

according to benefit in terms of stroke/SE, i.e., treatments which had the most pronounced impact in stroke/SE in underlying studies are reported first, while those which has the least pronounced impact occur last. We have also added this as a limitation to the Article Summary Section: "Icon arrays only consider the primary efficacy and safety endpoints of underlying studies and do not account for different clinical consequences associated with each of the outcomes".

4. If the authors insist on combining events of unequal weight, this should be explicitly discussed as limitation #1 and their rationale explained.

As discussed in Comment #3, we have not combined events of unequal weight. Our icon arrays provide both stroke/SE and major bleeding data in their disaggregated form (i.e., use different colors) which is not the case for composite outcomes. Further, at no point in our paper do we sum up the absolute risk results for stroke/SE and major bleeding which the reviewer is implying and is the case for composite outcomes. We simply summarize the findings from the primary and safety outcomes reported in underlying studies visually in disaggregated fashion using icon arrays. Nonetheless, we have revised the ordering of the icon arrays in the latest version of our manuscript according to benefit in terms of stroke, i.e., treatments which had the most pronounced impact in stroke/SE in underlying studies are reported first, while those which has the least pronounced impact occur last. We have also added a sentence to the article summary section highlighting this as a limitation.

5. Please move the data on edoxaban in the text (e.g., page 13), tables, and figures up so they are grouped with the other newer anticoagulants. Specifically, it makes more sense to discuss the safety results on page 13 starting with the drugs that have the least major bleeding and then proceeding with those that have more major bleeding. Right now, it looks like the data with edoxaban have just been appended at the end without consideration with where the data fit best.

We have made this change and modified all text, tables, figures in the main text and supplemental appendices accordingly. See comment #1

6. Please reconsider reporting on stroke + ICH without double-counting, by subtracting out the hemorrhagic strokes

See response to comment #3

7. Please reconsider some sort of weighting of the composite of efficacy and safety. Alternatively, you could modify the safety endpoint to only include more severe major bleeding events, as this would a more equitable composite of efficacy + safety (net outcome). Otherwise, this remains a major limitation of the analyses of net outcome that needs to be more explicitly discussed, with a rationale for why the analyses was performed in this way (in contrast to the approaches others have taken).

See response to comment #2, #3 and #4